# Hypoxia-activated neuropeptide Y/Y5 receptor/ RhoA pathway triggers chromosomal instability and bone metastasis in Ewing sarcoma

Congyi Lu[1,2,15], Akanksha Mahajan[3,15], Sung-Hyeok Hong[3,15], Susana Galli[3], Shiya Zhu[3], Jason U. Tilan [4], Nouran Abualsaud[3,5,6], Mina Adnani[3], Stacey Chung [4], Nada Elmansy[3], Jasmine Rodgers[3], Olga Rodriguez[7,8], Christopher Albanese[7,8], Hongkun Wang[9], Maureen Regan[10], Valerie Zgonc[11], Jan Blancato[7], Ewa Krawczyk[12], G. Ian Gallicano[3], Michael Girgis[7], Amrita Cheema[7], Ewa Iżycka-Świeszewska[13], Luciane R. Cavalli[7,14], Svetlana D. Pack[11] & Joanna Kitlinska[3✉]

Adverse prognosis in Ewing sarcoma (ES) is associated with the presence of metastases, particularly in bone, tumor hypoxia and chromosomal instability (CIN). Yet, a mechanistic link between these factors remains unknown. We demonstrate that in ES, tumor hypoxia selectively exacerbates bone metastasis. This process is triggered by hypoxia-induced stimulation of the neuropeptide Y (NPY)/Y5 receptor (Y5R) pathway, which leads to RhoA over-activation and cytokinesis failure. These mitotic defects result in the formation of polyploid ES cells, the progeny of which exhibit high CIN, an ability to invade and colonize bone, and a resistance to chemotherapy. Blocking Y5R in hypoxic ES tumors prevents polyploidization and bone metastasis. Our findings provide evidence for the role of the hypoxia-inducible NPY/Y5R/RhoA axis in promoting genomic changes and subsequent osseous dissemination in ES, and suggest that targeting this pathway may prevent CIN and disease progression in ES and other cancers rich in NPY and Y5R.

[1] Department of Physiology and Biophysics, Georgetown University, Washington, DC, United States. [2] New York Genome Center, New York, NY, United States. [3] Department of Biochemistry and Molecular & Cellular Biology, Georgetown University Medical Center, Washington, DC, United States. [4] Department of Human Science, School of Nursing and Health Studies, Georgetown University, Washington, DC, United States. [5] Cell Therapy and Cancer Research Department, King Abdullah International Medical Research Center, Riyadh, Saudi Arabia. [6] King Saud bin Abdulaziz University for Health Sciences, Riyadh, Saudi Arabia. [7] Department of Oncology, Lombardi Comprehensive Cancer Center, Georgetown University Medical Center, Washington, DC, United States. [8] Center for Translational Imaging, Georgetown University Medical Center, Washington, DC, United States. [9] Department of Biostatistics, Bioinformatics and Biomathematics, Georgetown University, Washington, DC, United States. [10] Genome Editing Core, University of Illinois, Chicago, Il, United States. [11] Laboratory of Pathology, National Cancer Institute, National Institutes of Health, Bethesda, MD, United States. [12] Center for Cell Reprogramming, Georgetown University Medical Center, Washington DC, United States. [13] Department of Pathology and Neuropathology, Medical University of Gdańsk, Gdańsk, Poland. [14] Research Institute Pelé Pequeno Príncipe, Faculdades Pequeno Príncipe, Curitiba, PR, Brazil. [15] These authors contributed equally: Congyi Lu, Akanksha Mahajan, Sung-Hyeok Hong. ✉email: jbk4@georgetown.edu

The presence of metastases is the most adverse prognostic factor in Ewing sarcoma (ES), an aggressive bone and soft tissue tumor affecting children and adolescents[1]. While patients with localized disease have a 5-year event-free survival (EFS) of 65-80%, a 3-year EFS for those with metastatic tumors remains below 30%, and decreases to 8–14% in patients with osseous dissemination[1–3]. This lack of effective therapies reflects a poor understanding of the mechanisms underlying ES metastasis. Although ES is triggered by chromosomal translocations that lead to the formation of EWS-ETS fusion proteins, the etiology of ES progression is unclear, as the same translocations are present in localized and metastatic tumors and the mutation rate of ES is low[1,4,5]. In addition, ES dissemination has been associated with decreased transcriptional activity of EWS-FLI1, suggesting that metastasis in these tumors is driven by mechanisms distinct from their initiation[6–8].

One of the few genetic markers of worse survival in ES is the presence of secondary chromosomal changes, including chromosome gains and complex karyotypes[1,9–12]. Such aberrations commonly arise due to mitotic defects leading to the formation of polyploid cells, which subsequently undergo atypical cell divisions with chromosome losses[13,14]. This, in turn, drives chromosomal instability (CIN), which facilitates cancer cell adaptation to stressors, such as therapies, hypoxia, or a metastatic niche microenvironment[13,14].

In ES, the chromosomal alterations consistent with CIN triggered by polyploidy correlate with chemoresistance, and their frequency increases in metastases and relapsing tumors[15–17]. Additionally, concurrent TP53 and STAG2 mutations, which may promote CIN, are associated with a dismal prognosis in ES, while STAG2 mutations alone correlate with metastasis[4,5,18]. Thus, even though ES is triggered by a single chromosomal translocation and is considered a genomically quiet tumor, CIN appears to contribute to its progression. Yet, the factors triggering CIN in ES remain unknown.

Hypoxia is another factor implicated in worse outcomes of ES. In ES patients, the presence of non-perfused tumor areas or necrosis correlates with adverse outcomes, multiple metastases and frequent bone dissemination[19,20]. In vitro, hypoxia increases the invasiveness of ES cells and up-regulates metastatic genes, while activation of HIF-1α promotes metastasis[21–25]. Yet, the mechanisms by which hypoxia exerts such effects are not fully understood.

Neuropeptide Y (NPY) and its receptors are expressed in various tumors and have multifaceted actions relevant to cancer biology, such as regulation of angiogenesis, cell proliferation, differentiation, survival, motility, and invasiveness[26–33]. We have shown that ES cells express high levels of NPY and its receptors, Y1R and Y5R, which is dependent on EWS-FLI1 transcriptional activity[27,34,35]. We have also established that hypoxia increases the expression of NPY and Y5R, as well as induces the expression of Y2R[23]. Moreover, hypoxia up-regulates dipeptidyl peptidase IV (DPPIV), a protease which cleaves NPY, converting it to a selective Y2R/Y5R agonist[23,36]. These changes in the NPY system shift its activity in ES cells to the Y2R/Y5R axis, which promotes their pro-metastatic features, such as cell motility[23]. However, since high constitutive expression of Y5R is driven by EWS-FLI1 and further increased by hypoxia, the levels of this receptor exceed those of inducible Y2R[23,27]. Hence, it is plausible that Y5R can also be activated independently of Y2R in hypoxic ES.

The goals of the current study were to test the effect of hypoxia on ES dissemination and determine the contribution of the NPY axis to this process. Our results identified hypoxia-induced over-activation of the NPY/Y5R/RhoA pathway as a mechanism leading to CIN and osseous dissemination. We have also shown that blocking Y5R in hypoxic ES tumors prevented hypoxia-induced genomic changes and bone metastasis. These findings provide the rationale for targeting Y5R in ES to prevent the disease progression.

## Results

**Hypoxia promotes ES osseous metastasis**. To test the effect of hypoxia on ES metastasis, we used an orthotopic ES xenograft model in combination with femoral artery ligation (FAL) to trigger tumor ischemia (Fig. 1a)[37]. ES cells were injected into gastrocnemius muscles of SCID/bg mice, in close proximity to the tibial crest to allow for investigation of their local bone invasiveness. Primary tumors were allowed to grow to a volume of 150–250 mm$^3$. Then, the tumor-bearing leg was either amputated (control) or subjected to FAL followed by amputation 3 days later. After amputation, metastasis was monitored by longitudinal MRI, followed by histopathological analysis. As previously described, this approach recapitulates ischemia and reoxygenation seen in solid tumors[37]. Control tumors at the size of 150–250 mm$^3$ had low endogenous hypoxia (Fig. S1a), while tumors from FAL-treated mice were severely hypoxic and necrotic, with only limited areas of viable tissue (Fig. S1b). In these areas, tumor cells that were previously hypoxic, as evidenced by positive staining for hypoxyprobe-1 (HP-1), exhibited signs of angioinvasive potential (Fig. S1c). The increase in hypoxia in FAL-treated ES xenografts was previously confirmed by us based on quantitative analysis[37].

The effect of hypoxia on ES dissemination was tested using two cell lines: SK-ES-1, which metastasizes to extrapulmonary locations, including bone and soft tissue, and TC71, which preferentially disseminates to lung[38]. SK-ES-1 xenografts in FAL-treated mice developed multiple metastases originating in bone, as seen by MRI and confirmed by histopathology (Fig. 1b). While FAL-induced hypoxia significantly exacerbated overall metastasis in mice with SK-ES-1 xenografts, this effect was due to the increase in osseous dissemination, without statistically significant differences in extra-osseous metastases. The FAL group had a decreased latency and an increased frequency of bone metastases (Fig. 1c, d), as well as a higher percentage of mice presenting with osseous lesions (Fig. 1e). Similarly, hypoxia-induced osseous dissemination of TC71 tumors, which typically do not form bone metastases, while extra-osseous metastases, including lung lesions, were unaffected (Fig. 1f, h)[38].

**Hypoxia induces aneuploidy in ES tumors and ES cells in vitro**. To determine the mechanisms of osseous dissemination following hypoxic stress, ES cell cultures were re-established from the above xenograft tissues that were harvested during amputation (primary tumors) and at euthanasia (metastases). While cells from small (150 mm$^3$) control SK-ES-1 primary tumors had nuclear sizes comparable to those in the original SK-ES-1 cell line, cells isolated from primary tumors of FAL-treated mice exhibited an increased frequency of hypertrophic cells with enlarged nuclei (Figs. 2a, and S2a, b). Similar abnormalities were seen in ES cells derived from large (1000 mm$^3$) control tumors, which exhibit high endogenous hypoxia (Figs. 2a and S2a, b)[23]. Flow cytometry confirmed that the nuclear hypertrophy observed in hypoxic ES cells was associated with a high prevalence of cells with DNA content exceeding that of diploid cells in the G2/M phase (>4c) (Fig. 2b). Moreover, FISH with probes for centromeres of chromosomes 3, 7, and 17, as well as the CDKNA2 gene locus on chromosome 9 (CDKNA2/CEN3/7/17) revealed an increased frequency of numerical chromosome aberrations in cells derived from primary tumors of FAL-treated mice and their corresponding metastases (Fig. 2c). Similarly, increases in the prevalence of ES cells with chromosomal gains were found in tissues from metastases arising

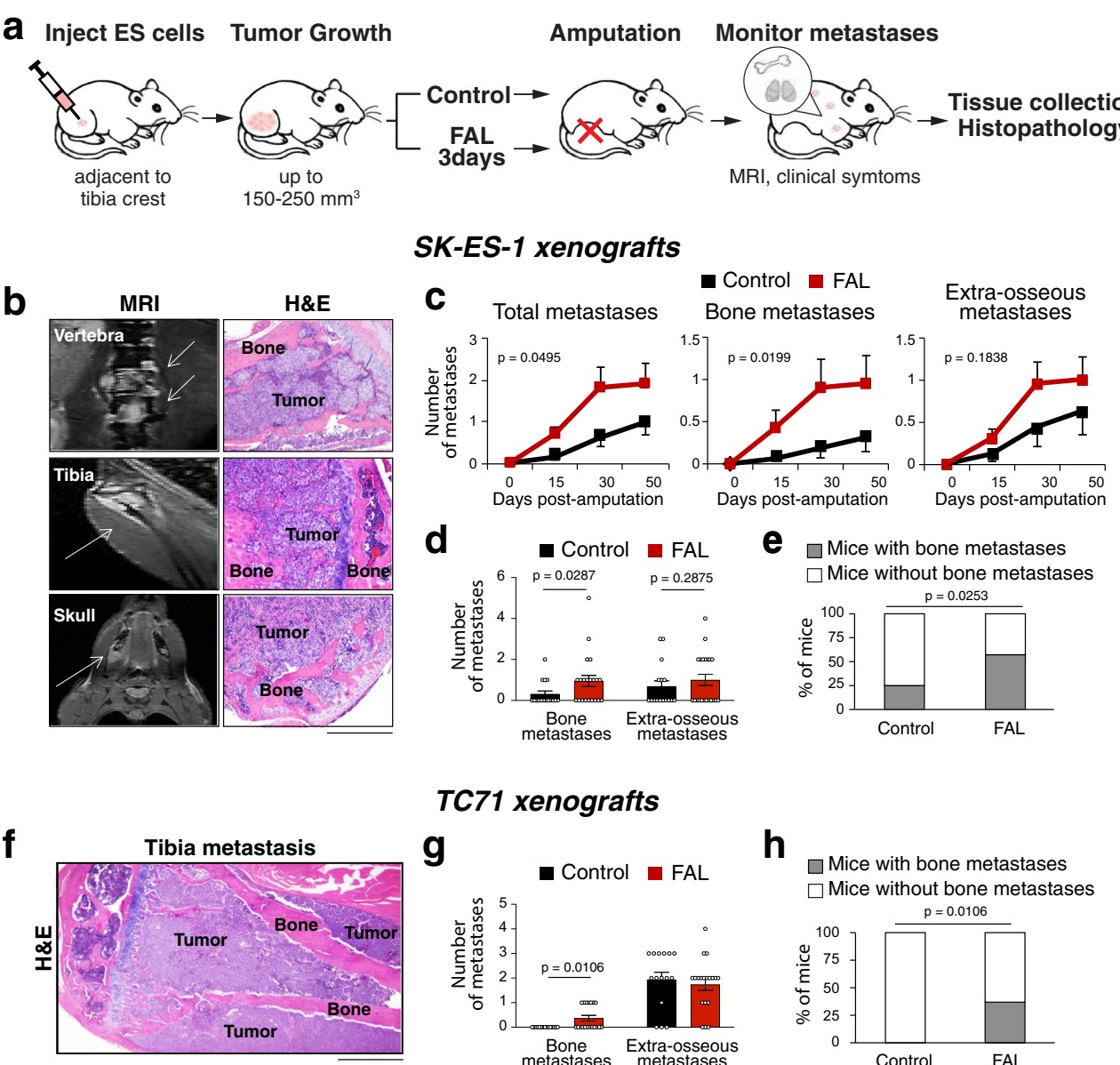

**Fig. 1 Hypoxia exacerbates osseous metastases in ES animal model. a** Design of the experiments testing the effect of tumor hypoxia triggered by femoral artery ligation (FAL) on ES metastasis. **b–e** SK-ES-1 xenografts: **b** Representative images of bone metastases detected by MRI and confirmed by histopathology (H&E) ($n = 20$). Scale bar: 400 μm. **c** Time line of metastasis from control or hypoxic xenografts. Hypoxia effects assessed by the generalized estimating equation (GEE) test. **d** The comparison of bone and extra-osseous metastases in control and FAL-treated animals detected at the end point of the experiment. One-tailed Mann–Whitney test. **e** Percentage of mice with bone metastases in control and hypoxic groups. One-sided $\chi^2$ test. For panels **c–e**, $n = 16$ and 21 for control and FAL groups, respectively. **f–h** TC71 xenografts: **f** Representative image of bone metastasis detected by histological analysis ($n = 7$). Scale bar: 1000 μm. **g** The analysis of bone and extra-osseous metastases in control or FAL-treated mice. Two-tailed Mann–Whitney test. **h** Percentage of mice with bone metastases in control and hypoxic groups. Two-sided Fisher's exact test. For panels **g**, **h**, $n = 15$ and 19 for control and FAL groups, respectively. Error bars indicate standard error of the mean.

in FAL-treated animals, validating the above observations in vitro (Fig. S2c, d). Lastly, the cell lines derived from hypoxic primary tumors and their corresponding metastases exhibited an increased frequency of mitotic segregation errors and abnormalities in interphase nuclei that are markers of chromosomal instability (Figs. 2d and S2e, f).

Since genomic changes in hypoxic ES tumors can be triggered by a variety of factors activated by ischemia, we tested whether hypoxia in vitro recapitulates this effect. Culture of SK-ES-1 cells in 0.1% oxygen for 72 h increased the occurrence of cells with enlarged nuclei (Fig. 2e). This was preceded by the increase in the frequency of multinucleated cells observed upon 24 h hypoxia

exposure, implicating defects in cytokinesis as the mechanism leading to polyploidization in hypoxic ES cells (Fig. 2f). A parallel experiment performed on the co-culture of SK-ES-1 cells expressing GFP or mCherry exposed to hypoxia for 24 h revealed a lack of multinucleated cells expressing both fluorescent proteins, thereby excluding the contribution of cell fusions to their formation (Fig. S3a). Flow cytometry demonstrated an increased percentage of cells with >4c DNA content upon hypoxia (Fig. 2g), while FISH with centromeric probes confirmed the presence of chromosome gains (Fig. 2h). Stable transfection of SK-ES-1 cells with the FUCCI Cell Cycle Sensor revealed that proliferation upon 72 h exposure to severe hypoxia was almost

## Cells cultured from hypoxic SK-ES-1 xenografts

## SK-ES-1 cells subjected to hypoxia in vitro

**Fig. 2 Hypoxia leads to the accumulation of cells with increased DNA content. a** Representative images of cells cultured from control or FAL-treated SK-ES-1 xenografts stained with DAPI and wheat germ agglutinin (WGA), followed by the analysis of nuclear area of the original SK-ES-1 cells ($n = 200$) and cells isolated from small (150 mm$^3$) control ($n = 417$) or FAL-treated tumors ($n = 431$), or large (1000 mm$^3$) untreated primary tumors ($n = 405$). Cells cultured from xenografts were derived from 10 tumors per group. Scale bar: 100 μm. One-way ANOVA followed by Dunnett's test. **b** Flow cytometry analysis of DNA content in cells cultured from control or FAL-treated SK-ES-1 xenografts, followed by the analysis of cells exceeding 4c DNA ($n = 3$ tumors/group). One-side t-test; **c** Representative images of FISH with CDKNA2/CEN3/7/17 probes in the original SK-ES-1 cells and cells cultured from SK-ES-1 primary tumors or metastases of FAL-treated mice, followed by the analysis of numerical chromosome alterations (Control primary tumors $n = 90$ cells from 7 tumors, FAL primary tumors $n = 160$ cells from 6 tumors, FAL metastases $n = 45$ cells from 2 tumors). Two-sided Fisher's exact test. **d** Mitotic segregation errors assessed in the above cells (Control primary tumors $n = 24$, FAL primary tumors $n = 33$, FAL metastases $n = 21$ cells). Two-sided Fisher's exact test. Scale bar: 10 μm. **e** Representative images of SK-ES-1 cells cultured in normoxia (NOR) or 0.1% oxygen (Hypoxia, HYP) for 24 h or 72 h stained with DAPI and WGA, followed by the analysis of nuclear area (NOR $n = 351$, HYP 24 h $n = 292$, HYP 72 h $n = 385$). Scale bar: 100 μm. One-way ANOVA followed by Dunnett's test. **f** Frequency of multinucleated and hypertrophic SK-ES-1 cells upon exposure to HYP ($n = 3$ independent experiments). One-way ANOVA followed by Dunnett's test. **g** Representative flow cytometry quantification of DNA content in NOR or HYP SK-ES-1 cells, followed by the analysis of cells with >4c DNA ($n = 3$ independent experiments). One-sided paired t-test. **h** Representative image of FISH with CDKNA2/CEN3/7/17 probes in SK-ES-1 cells cultured in HYP for 72 h ($n = 189$ cells). Scale bar: 10 μm. **i** Analysis of mitotic segregation errors in SK-ES-1 cells - control or subjected to HYP for 72 h followed by 24 h culture in NOR ($n = 24$ and 38 cells per group, respectively). Two-sided Fisher's exact test. **a**, **e**: H – hypertrophic cells; M – multinucleated cells; **c**, **h**: green borderlines – chromosome gains. Error bars indicate standard error of the mean. For violin plots in **a** and **e**, the red lines represent the median; the black lines represent the quartiles.

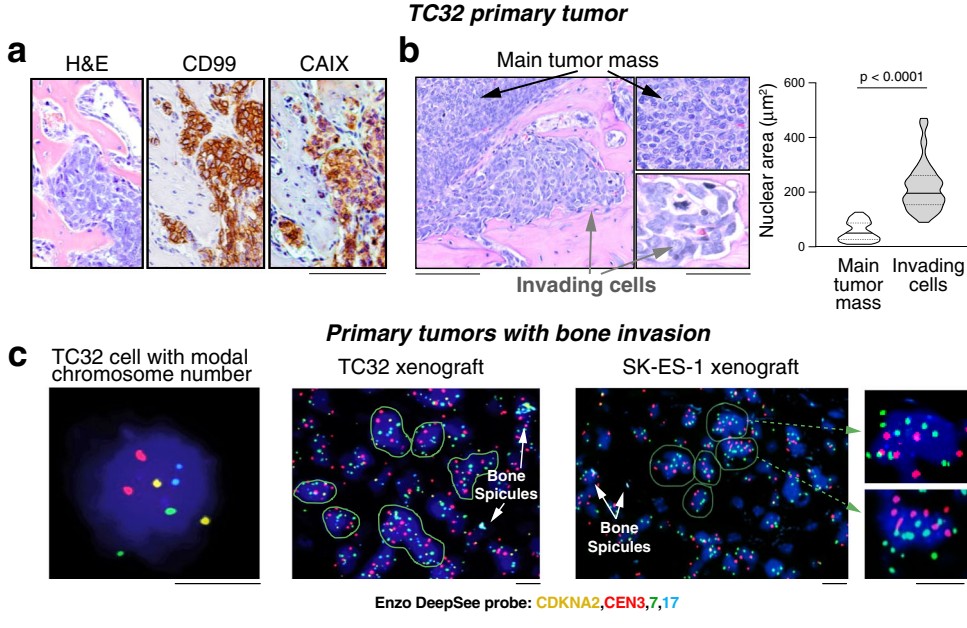

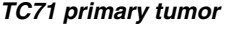

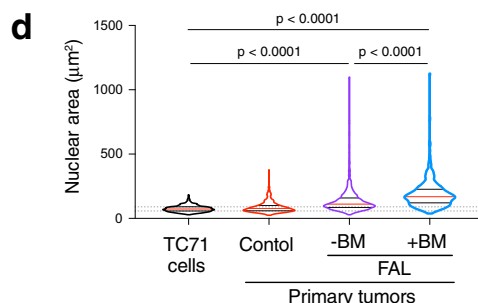

**Fig. 3 The presence of hypertrophic cells accompanies bone colonization. a** Representative images of TC32 primary tumor showing tumor cells growing in bone invasion areas, stained with H&E or immunostained for the ES marker, CD99, and a hypoxia marker, carbonic anhydrase IX (CAIX) ($n = 5$). Scale bar: 100 μm. **b** Representative images of TC32 cells growing in main tumor mass and in areas of local bone invasion, followed by the analysis of nuclear area ($n = 54$ from three independent tumors per group). The solid lines represent the median; the dashed lines represent the quartiles. Scale bars: 100 μm or 40 μm (insets). Two-tailed *t*-test. Error bars indicate standard error of the mean. **c** FISH with CDKNA2/CEN3/7/17 probes performed in bone invasion areas of TC32 and SK-ES-1 xenografts ($n = 16$ for TC32 xenografts, $n = 6$ for SK-ES-1 xenografts). Scale bar: 10 μm. Green borderlines – cells with chromosome gains. **d** Violin plot shows the analysis of nuclear area of cell cultured from TC71 control or FAL-treated xenografts, which did (+BM) or did not (-BM) metastasize to bone (original TC71 cells $n = 285$, control $n = 401$ cells from 8 tumors, FAL –BM $n = 366$ cells from 7 tumors, and FAL + BM $n = 360$ cells from 4 tumors). The red lines represent the median; the black lines represent the quartiles. One-way ANOVA followed by Tukey's test.

exclusively limited to hypertrophic cells (Fig. S3b), while subsequent stimulation of cell divisions resulted in frequent mitotic segregation errors (Fig. 2i). Glucose remained readily available in the hypoxic cell culture media throughout the duration of the experiment, eliminating nutrient deficiency as a potential trigger of the above effects (Fig. S3c). These data implicate hypoxia as a key factor in the induction of polyploidy and subsequent CIN in ES cells.

**The presence of hypertrophic ES cells accompanies bone colonization**. In addition to the overall increased frequency of enlarged cells in hypoxic ES xenografts, we observed their accumulation in areas of local bone invasion in control tumors (Fig. 3). This was found in SK-ES-1 and TC32 xenografts, both of which have a high ability to invade bone[38]. ES cells located in the invaded niches within bone were positive for the ES marker, CD99, and a hypoxia marker, carbonic anhydrase IX (CAIX) (Figs. 3a and S4a)[39]. In addition, these cells had enlarged nuclei, as compared to the cells in the main tumor mass outside of the

bone (Fig. 3b). FISH with CDKNA2/CEN3/7/17 probes performed on tumor tissues from TC32 and SK-ES-1 xenografts revealed frequent chromosome gains in these cells (Fig. 3c). Collectively, this data suggests that the hypoxia that developed in areas of bone invasion triggered polyploidy. TC71 primary tumors have a low ability to invade bone tissue locally[38]. However, within the TC71 FAL group, xenografts that gained the capacity to form distant bone metastases exhibited a higher prevalence of cells with enlarged nuclei, as compared to tumors that did not target bone (Figs. 3d and S4b, c), indicating an association between the degree of hypoxia-induced polyploidy and osseous dissemination.

**The progeny of hypoxia-induced polyploid ES cells preferentially metastasize to bone**. To test the metastatic properties of polyploid ES cells arising in hypoxia, we isolated diploid and tetraploid cell fractions from normoxic or hypoxic SK-ES-1 cultures, the progeny of which was subsequently used in an orthotopic xenograft model (Fig. 4a). To this end, we employed the

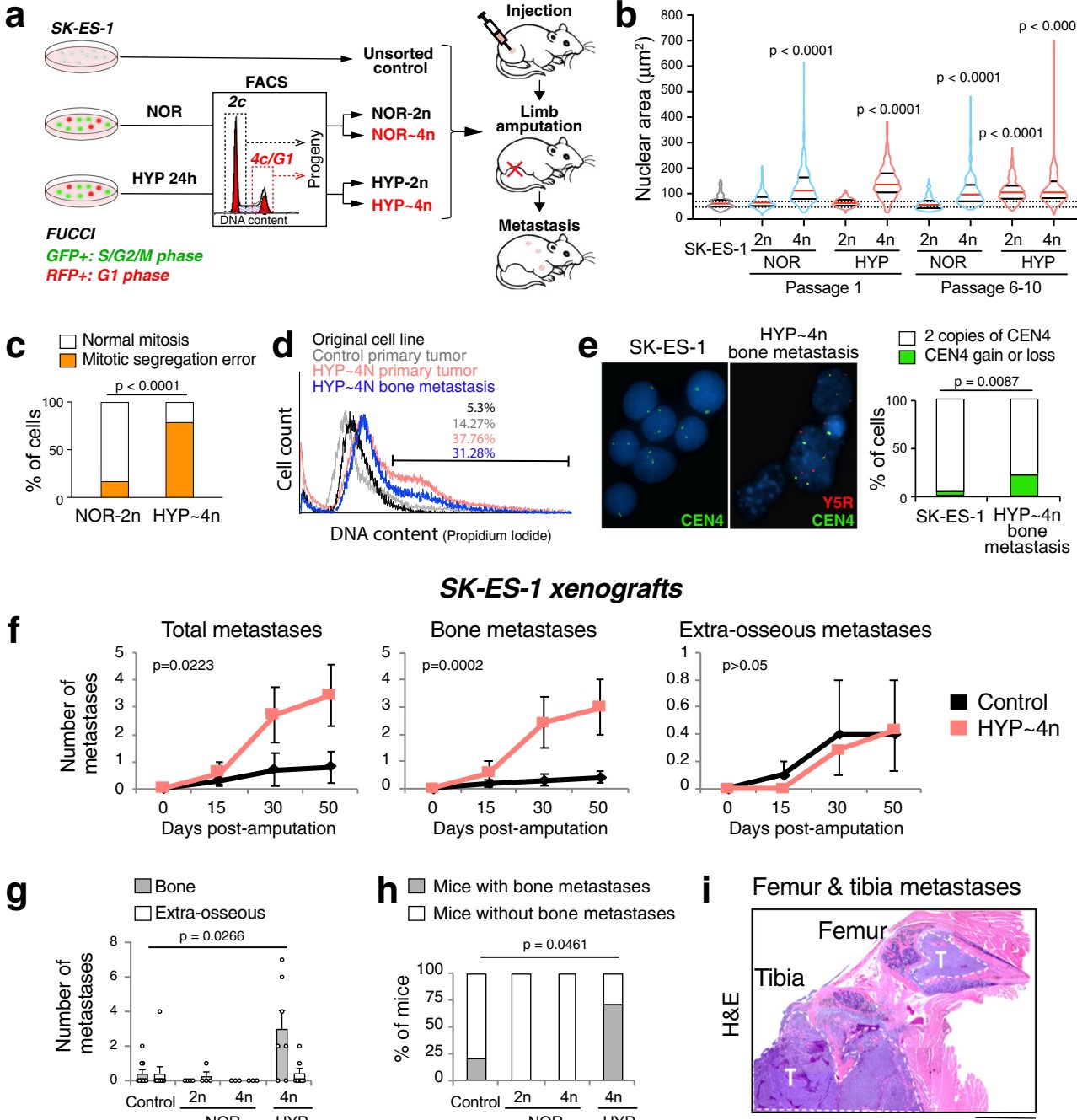

**Fig. 4 The progeny of hypoxia-induced polyploid cells preferentially metastasize to bone. a** Design of the experiments for investigating the metastatic properties of the progeny of diploid (2n) and teraploid (4n) cell fractions isolated from normoxic (NOR) or hypoxic (HYP) SK-ES-1 cells using the FUCCI Cell Cycle Sensor followed by DNA staining and FACS. **b** Violin plot shows the analysis of nuclear area of the isolated diploid and tetraploid populations from NOR or HYP cells at passages 1 or 6–10 after sorting (Control SK-ES-1 $n = 137$; passage 1: NOR-2n $n = 384$, NOR~4n $n = 282$, HYP-2n $n = 334$, HYP~4n $n = 143$; passage 6–10: NOR-2n $n = 161$, NOR~4n $n = 132$, HYP-2n $n = 173$, HYP~4n $n = 162$). The red lines represent the median; the black lines represent the quartiles. $P < 0.0001$ vs. SK-ES-1 cells; one-way ANOVA followed by Dunnett's test. **c** Analysis of mitotic segregation errors in NOR-2n ($n = 24$) or HYP~4n SK-ES-1 cells ($n = 38$). Two-sided Fisher's exact test. **d** Flow cytometry analysis of DNA content in the original cell line and the cells cultured from parental and HYP~4n SK-ES-1 xenografts. **e** Representative images of FISH with CEN4/Y5R probes in the original SK-ES-1 cells ($n = 53$) or cells cultured from HYP~4n bone metastasis ($n = 168$), followed by the analysis of CEN4 numerical aberrations. Two-sided Fisher's exact test. Scale bar: 10 μm. **f** Analysis of metastasis development in mice with control ($n = 10$) or HYP~4n ($n = 7$) SK-ES-1 xenografts; Generalized estimating equation (GEE) test. **g, h** Analysis of number of metastases (**g**) and the percentage of animals with bone metastases (**h**) in mice with SK-ES-1 xenografts developed from control ($n = 10$) or sorted cell fractions (NOR-2n $n = 4$, NOR-4n $n = 3$, HYP~4n $n = 7$). Error bars indicate standard error of the mean. Two-tailed Mann–Whitney test (**g**); One-sided $\chi^2$ test (**h**). **i** Representative image of multiple HYP~4n osseous metastases in the hind limb of mice bearing HYP~4n SK-ES-1 xenografts ($n = 5$). Scale bar: 2000 μm. T - tumor.

FUCCI Cell Cycle Sensor followed by cell sorting based on the DNA content in order to separate tetraploid cells, i.e. cells with 4c DNA in G1phase of cell cycle, from dividing diploid cells in G2/M phase[40]. The subsequent propagation of these cell fractions allowed for obtaining their progeny: diploid populations from normoxic and hypoxic conditions termed NOR-2n and HYP-2n, respectively, and near-tetraploid cells arising due to aberrant divisions of 4n cells, termed NOR~4n and HYP~4n (Fig. 4a). Analysis of nuclear size performed immediately after sorting confirmed that the NOR-2n and HYP-2n cells had nuclei comparable to the unsorted SK-ES-1 cells, while the NOR-~4n and HYP~4n cells had enlarged nuclei (Fig. 4b). However, after 6–10 passages in vitro under normoxic conditions, the HYP-2n cells had enlarged nuclei, resembling HYP~4n cells (Fig. 4b). No such shift in nuclear sizes was observed in NOR-2n cells. Hence, hypoxia induced durable effects that resulted in the formation of polyploid cells upon reoxygenation. HYP~4n cells exhibited frequent mitotic segregation errors indicating their CIN (Fig. 4c).

Primary tumors arising from all tested cell fractions had similar growth rates (Fig. S5a). However, HYP~4n xenografts presented with a more aggressive phenotype, as compared to control SK-ES-1 tumors. Cells isolated from HYP~4n primary tumors and metastases exhibited signs of CIN, which was indicated by increased DNA content, variable nuclear size and chromosome number alterations (Fig. 4d, e). Mice bearing HYP~4n xenografts had an increased number of total metastases caused by the higher frequency of bone lesions, while extra-osseous metastases remained unchanged (Fig. 4f). Bone was the main metastatic site of HYP~4n xenografts (87.5% of metastases), with multiple bone lesions per animal (Fig. 4g–i). This phenotype resembled the metastatic pattern of FAL-treated mice. No bone metastases were detected in mice with NOR-2n and NOR~4n xenografts (Fig. 4g, h). Aside from their capability to disseminate to bone, HYP~4n cells were resistant to doxorubicin, suggesting a role for polyploidy in the failure of chemotherapy in patients with bone metastasis (Fig. S5b, c).

**Y5R overexpression induces polyploidy due to defects in cytokinesis**. ES cells constitutively express NPY and its receptor, Y5R (Fig. S6a)[27]. Expression of both genes is further elevated in hypoxia, as previously demonstrated in a panel of ES cell lines[23], and shown here for Y5R in SK-ES-1 and TC71 cells (Fig. S6b, c). To determine the impact of Y5R induction on ploidy, CHO-K1 cells, which express negligible levels of NPY and its receptors, were stably transfected with Y5R-EGFP cDNA[41]. The expression of Y5R in CHO-K1/Y5R-EGFP cells was comparable to its endogenous levels in ES cells (Fig. S7a). Co-localization of the Y5R-EGFP fusion protein with a plasma membrane marker, wheat germ agglutinin (WGA), confirmed its membrane localization (Fig. S7b). Notably, CHO-K1/Y5R-EGFP stable transfectants became hypertrophic, with enlarged or multiple nuclei, while no such aberrations were observed after transfection with other NPY receptors or with EGFP alone (Fig. 5a). The hypertrophic CHO-K1/Y5R-EGFP cells were actively proliferating, as indicated by positive staining for EdU, confirming that the observed changes were not associated with cell senescence (Fig. S7c). A similar phenotype was observed in several independently developed clones transfected with either human or rat Y5R cDNA. The CHO-K1/Y5R-EGFP cells exhibited increased DNA content and chromosome modal numbers (Fig. 5b, c).

To identify the mechanism leading to aneuploidy in the CHO-K1/Y5R transfectants, we examined early events following induction of Y5R expression by transient transfection with Y5R cDNA. Using time-lapse microscopy, we established that the transfection of CHO-K1 cells with EGFP, Y1R-EGFP or Y2R-EGFP had no apparent effect on cell divisions, which were

completed within 3–4 h from the time when mitotic cell rounding was first observed (Sup. movie 1). Conversely, the Y5R-EGFP positive cells exhibited defects in detachment of daughter cells at the last stage of cytokinesis. This arrest lasted for 10-20 h and often resulted in cell death (Fig. 5d and Sup. movie 2). Nevertheless, fusions of the daughter cells were also observed, leading to the formation of hypertrophic cells with enlarged nuclei (Fig. 5e and Sup. movie 3).

In agreement with the block in cytokinesis, NPY stimulation of CHO-K1/Y5R-EGFP cells resulted in p44/42 MAPK activation (Fig. 5f) and an increase in DNA synthesis, as measured by $^3$H-thymidine uptake, while the number of viable cells measured indirectly by MTS assay or by direct cell count did not change (Figs. 5g and S7d). Therefore, NPY initiates cell cycle progression in CHO-K1/Y5R-EGFP cells, as previously shown for Y1R and Y2R NPY receptors[41]. Yet, due to a subsequent block in cytokinesis, this process resembles endoreplication.

**Y5R-induced cytokinesis defects are mediated through RhoA over-activation**. The mitotic abnormalities observed in Y5R transfectants were consistent with defects in the abscission phase of cytokinesis. Such aberrations can be caused by sustained activation of RhoA, a key cytoskeleton regulator (Fig. 6a)[42]. Indeed, RhoA pull-down assay revealed increased levels of RhoA-GTP in CHO-K1/Y5R-EGFP cells treated with NPY, while the Y5R antagonist, CGP71683, decreased RhoA activity (Fig. 6b). Both Y5R and active RhoA accumulated in the cleavage furrow of CHO-K1/Y5R-EGFP cells at the abscission phase, a phenomenon not observed in non-transfected CHO-K1 cells (Fig. 6c). To directly test the role of RhoA in Y5R-induced formation of polyploid cells, CHO-K1 cells were transiently transfected with Y5R-EGFP cDNA in the presence or absence of the Rho inhibitor I, C3 Transferase, at a dose that was sufficient to block Y5R-induced RhoA activation, but did not interfere by itself with cell proliferation (Fig. S8). In media supplemented with 10% FBS, which contains NPY, Y5R-overexpression led to the formation of bi-nucleated cells 24 h after transfection and Rho inhibition prevented this effect (Fig. 6d). The number of multinucleated cells in medium deprived of NPY (0.1% FBS) was comparable to that observed in CHO-K1/EGFP control, while the exogenous peptide increased their frequency in a Rho-dependent manner (Fig. 6d). The above data established that cytokinesis defects triggered by NPY/Y5R axis signaling are the result of RhoA activation.

To determine whether the signaling pathways activated in CHO-K1/Y5R-EGFP cells are relevant to ES, we treated SK-ES-1 cells with selective Y1R, Y2R, or Y5R agonists. The Y5R agonist, BWX 46, significantly increased RhoA-GTP levels, while the Y1R and Y2R agonists failed to elicit this response (Fig. 6e). RhoA activity was also elevated in hypoxia and this effect was blocked by the Y5R antagonist (Fig. 6f). Consequently, blocking Y5R prevented hypoxia-induced nuclear hypertrophy, mitotic segregation errors and aneuploidy (Figs. 6g, h and S9).

**Blocking Y5R reduces ES bone metastasis**. To test the role of the NPY/Y5R axis in ES metastasis, mice-bearing SK-ES-1 orthotopic xenografts were treated with either vehicle or the Y5R antagonist, CGP71683. Treatments commenced when tumors reached 150 mm$^3$ and endogenous hypoxia was low, and continued for 5 days (Fig. S10a)[37]. During this period, tumors reached ~600 mm$^3$, and HP-1 staining confirmed the presence of hypoxic tumor tissue (Fig. S10b). In line with our in vitro findings, the Y5R antagonist reduced the prevalence of cells with enlarged nuclei in the primary tumors (Fig. 7a and S10c, d). While this treatment did not affect primary tumor growth (Fig. S10e) or extra-osseous metastases, it decreased the percentage of mice with

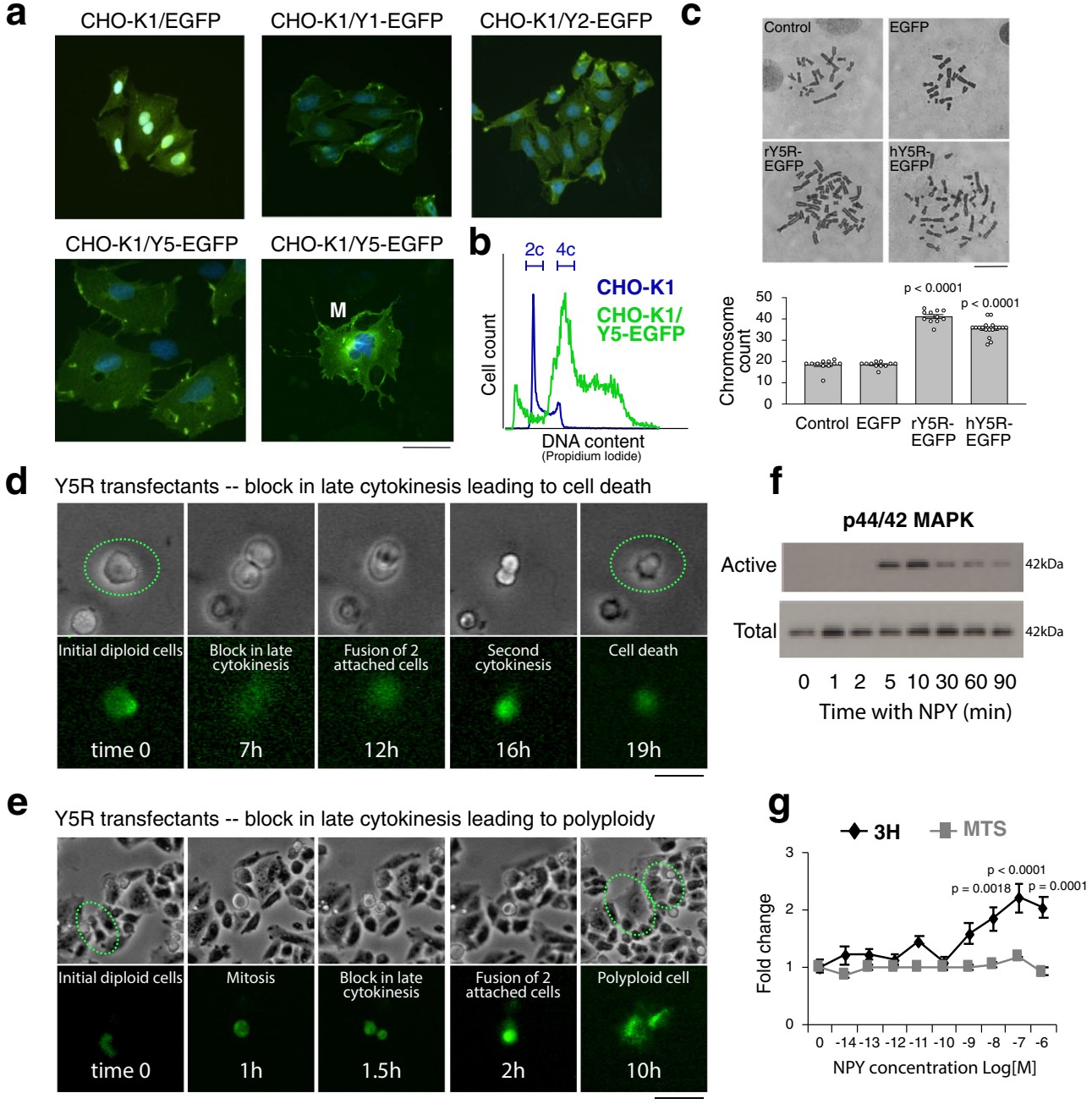

**Fig. 5 Over-expression of NPY Y5R leads to aneuploidy caused by cytokinesis defects. a** Representative fluorescence images of CHO-K1 cells transfected with EGFP alone or fused to the NPY receptors. M – multinucleated cell ($n = 3$–10 clones per group). Scale bar: 100 μm. **b** Flow cytometry analysis of DNA content in CHO-K1 cells transfected with Y5R-EGFP compared to non-transfected control. **c** Representative images of metaphase chromosomes from CHO-K1 cells (control) and CHO-K1 cells transfected with EGFP alone, rat Y5R-EGFP (rY5R-EGFP), or human Y5R-EGFP (hY5R-EGFP), followed by the analysis of chromosome counts ($n = 11$ for Control, EGFP and rY5R-EGFP, and $n = 18$ for hY5R EGFP). Scale bar: 50 μm. $P < 0.001$ vs control cells; Two-sided $t$-test. **d-e** Representative images from time lapse microscopy showing cytokinesis defects in CHO-K1/Y5R-EGFP transient transfectants, leading either to cell death (**d**) or polyploidy (**e**) ($n = 9$ per group). Scale bars: 20 μm (**d**) or 40 μm (**e**). **f** Time course of p44/42 MAPK activation upon stimulation with $10^{-7}$M NPY in CHO-K1/Y5R-EGFP stable transfectants detected by western blot ($n = 3$). **g** $^3$H-thymidine incorporation and MTS assay in CHO-K1/Y5R-EGFP stable transfectants following treatment with increasing NPY concentrations for 24 h ($n = 6$). One-way ANOVA followed by Dunnett's test. Error bars indicate standard error of the mean.

secondary bone lesions and the number of osseous metastases, although the latter difference did not reach statistical significance (Fig. 7b–d).

To further evaluate the role of Y5R in ES metastasis, we used the CRISPR/Cas9 system to knock-down its expression in ES cells. SK-ES-1 cells were stably transfected with doxycycline (Dox)-inducible Cas9 nuclease and sgRNA targeting *NPY5R* gene (SK-ES-1/Dox-Cas9/Y5R-sgRNA), and then tested in the xeno-graft model with or without FAL. In primary tumors, in vivo Dox administration (+Dox diet) induced *NPY5R* gene editing with approximately 65% efficiency, which resulted in a heterogeneous ES cell population (Fig. S11a, b). These changes did not affect

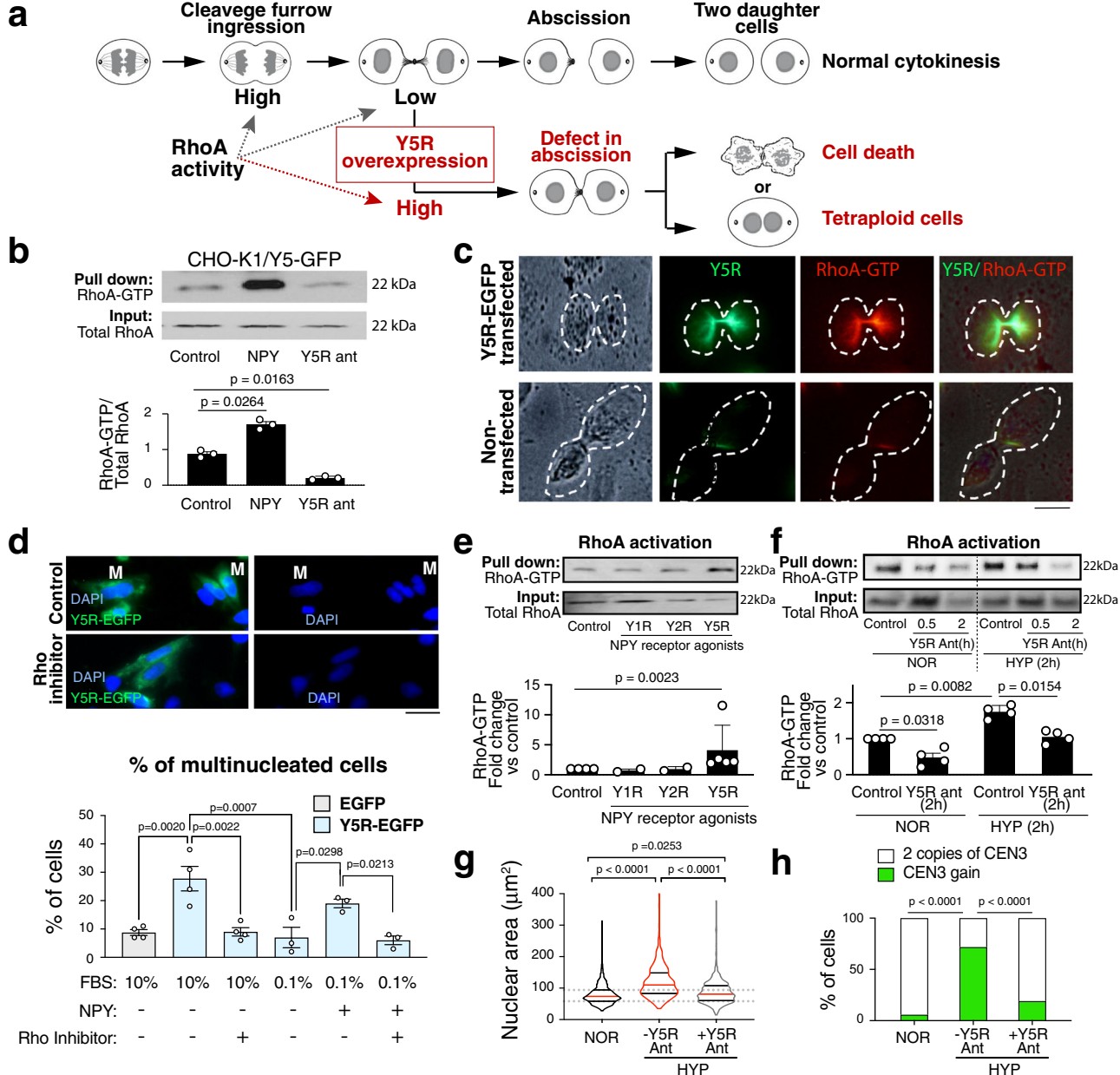

**Fig. 6 Y5R-induced cytokinesis defects are caused by aberrant RhoA activation. a** Sustained high RhoA activity during cytokinesis can lead to cytokinesis failure. **b** RhoA activity measured by a RhoA pull-down assay in CHO-K1 cells stably transfected with Y5R-EGFP. The cells were treated with $10^{-7}$M NPY for 20 min or $10^{-6}$M Y5R antagonist (ant), CGP 71683, for 30 min. Band intensities quantified by densitometry ($n = 3$). Two-tailed paired $t$-test. **c** Representative fluorescence images of CHO-K1 cells in late cytokinesis, control or transiently transfected with Y5R-EGFP, immunostained for Y5R and active RhoA-GTP ($n = 3$ per group). Scale bar: 10 µm. **d** Representative images of CHO-K1 cells 24 h after transfection with Y5R-EGFP, with ($+$) or without ($-$) Rho inhibitor I (0.01 µg/ml). The treatment was performed in the media supplemented with 10% or 0.1% FBS, with or without exogenous NPY ($10^{-7}$M). The cells were stained with DAPI, and used to quantify the prevalence of multinucleated cells (M) ($n = 4$ and 3 independent experiments for 10% and 0.1% FBS, respectively). Scale bar: 10 µm. One-way ANOVA followed by Tukey's test. **e** RhoA activity in SK-ES-1 cells treated with selective NPY receptor agonists ($10^{-7}$M for 20 min) measured by RhoA pull-down assay ($n = 5$). Two-tailed paired t-test. **f** RhoA activity measured by pull-down assay in SK-ES-1 cells cultured in normoxia (NOR) or subjected to hypoxia (HYP; 0.1% oxygen) for 2 h with or without Y5R antagonist ($10^{-6}$M) ($n = 4$). **e, f** Two-tailed paired t-test. **g** Violin plot shows the analysis of nuclear area of SK-ES-1 cells in NOR ($n = 419$) or HYP for 72 h, with Y5R antagonist ($10^{-6}$M) ($n = 420$) or without it ($n = 373$). Data from three independent experiments. The red lines represent the median; the black lines represent the quartiles. One-way ANOVA followed by Tukey's test. **h** Analysis of frequency of Chr.3 gains detected by FISH with CDKNA2/CEN3/7/17 probes in SK-ES-1 cells in NOR ($n = 137$) or HYP for 72 h, with Y5R antagonist ($10^{-6}$M) ($n = 115$) or without it ($n = 110$). Data from three independent experiments. Two-sided Fisher's exact test. Error bars indicate standard error of the mean.

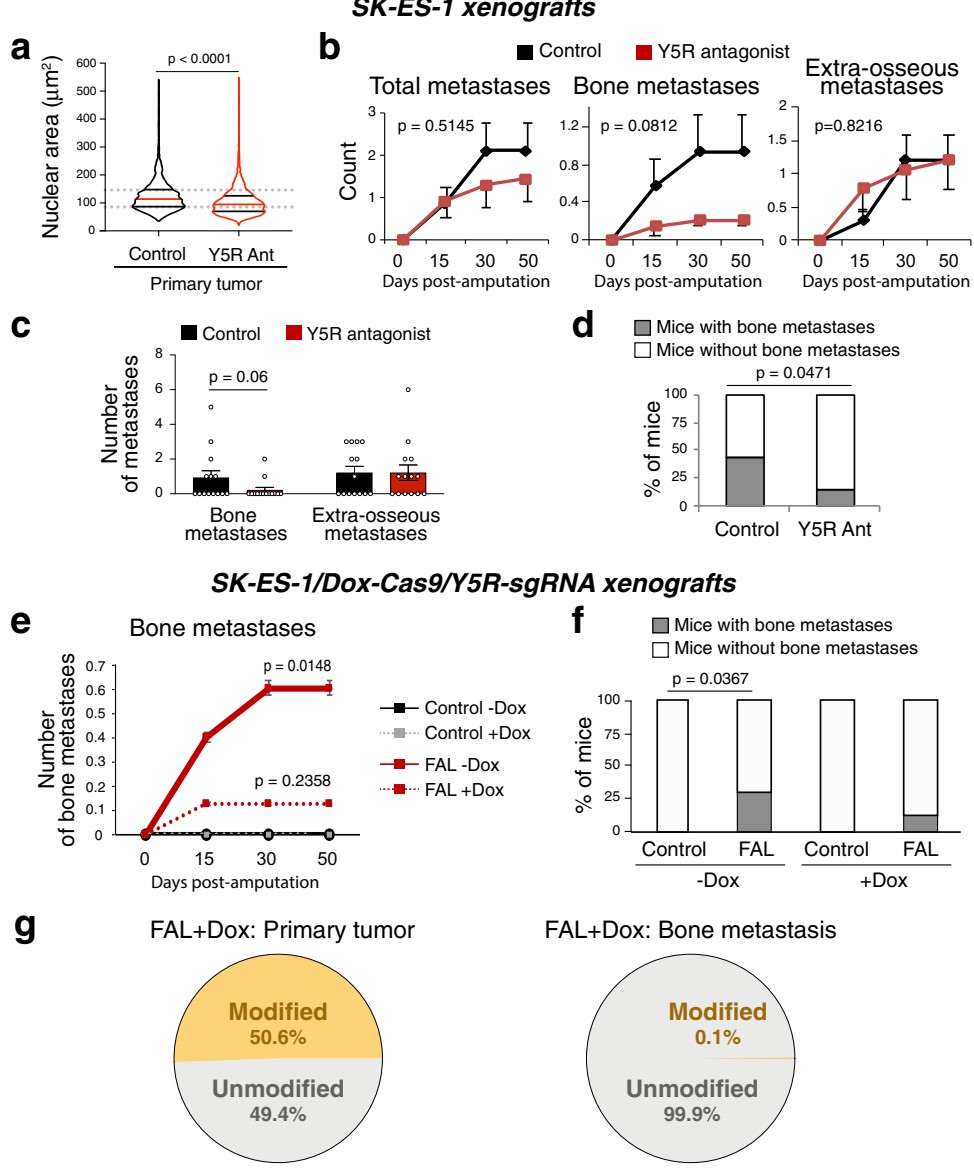

**Fig. 7 Y5R inhibition prevents bone metastasis in ES xenografts. a** Violin plot shows the analysis of nuclear area of cells cultured from control and Y5R antagonist-treated SK-ES-1 xenografts (CGP 71683, 20 mg/kg, 5 days) ($n = 2161$ cells from 10 tumors and $n = 2184$ cells from 12 tumors, respectively). The red lines represent the median; the black lines represent the quartiles. Two-tailed unpaired t-test. **b** Time course of SK-ES-1 metastasis development in control and Y5R-antagonist-treated mice. Generalized estimating equation (GEE) test. **c** Analysis of number of metastases in control and Y5R antagonist-treated mice. One-sided Mann–Whitney test. **d** Analysis of percentage of mice with osseous dissemination in control and Y5R antagonist-treated groups. One-sided $\chi^2$ test. (For panels **b**–**d**, $n = 14$ mice per group). **e** Time course of bone metastasis development in mice with SK-ES-1 xenografts transduced with doxycycline (Dox)-inducible Cas9 and Y5R sgRNA, on control (-Dox) or Dox-supplemented diet (+Dox), with or without FAL-induced tumor hypoxia. $P$ values as indicated compared to Control –Dox; GEE test. **f** The analysis of the percentage of mice with bone metastases in the experimental groups from e. One-sided $\chi^2$ test. (For panels e-f: Control –Dox $n = 9$, FAL -Dox $n = 10$, Control +Dox $n = 9$ and FAL + Dox $n = 8$ mice per group). **g** Frequency of *NPY5R* gene modifications in a primary tumor from a FAL-treated mouse on +Dox diet and in its corresponding bone metastasis. Error bars indicate standard error of the mean.

primary tumor growth (Fig. S11c). FAL-induced tumor hypoxia exacerbated osseous dissemination in SK-ES-1/Dox-Cas9/Y5R-sgRNA mice fed with control diet (-Dox), as evidenced by significant increases in the number of bone metastases (Fig. 7e) and the frequency of mice with secondary osseous lesions (Fig. 7f). Conversely, no statistically significant increase in bone metastasis was observed in FAL-treated mice on +Dox diet (Fig. 7e, f). Out of eight animals in this group, only one developed bone metastases, which were detected in two independent locations. Notably, one of the metastases retained an intact *NPY5R* sequence, suggesting that metastasis was initiated by Y5R-

positive ES clones (Fig. 7g). An insufficient amount of tissue was recovered from the second lesion to allow for sequencing to confirm this phenomenon. Dox treatment did not affect primary tumor growth and bone metastasis in wild type SK-ES-1 xenografts (Fig. S12). Hence, both pharmacological and genetic Y5R inhibition confirmed the role of the NPY/Y5R axis in hypoxia-induced bone metastasis.

**Relevance to human ES and other tumors.** To provide direct evidence for the presence of hypertrophic cells in hypoxic tumors, we performed histopathological analysis of tissues from ES

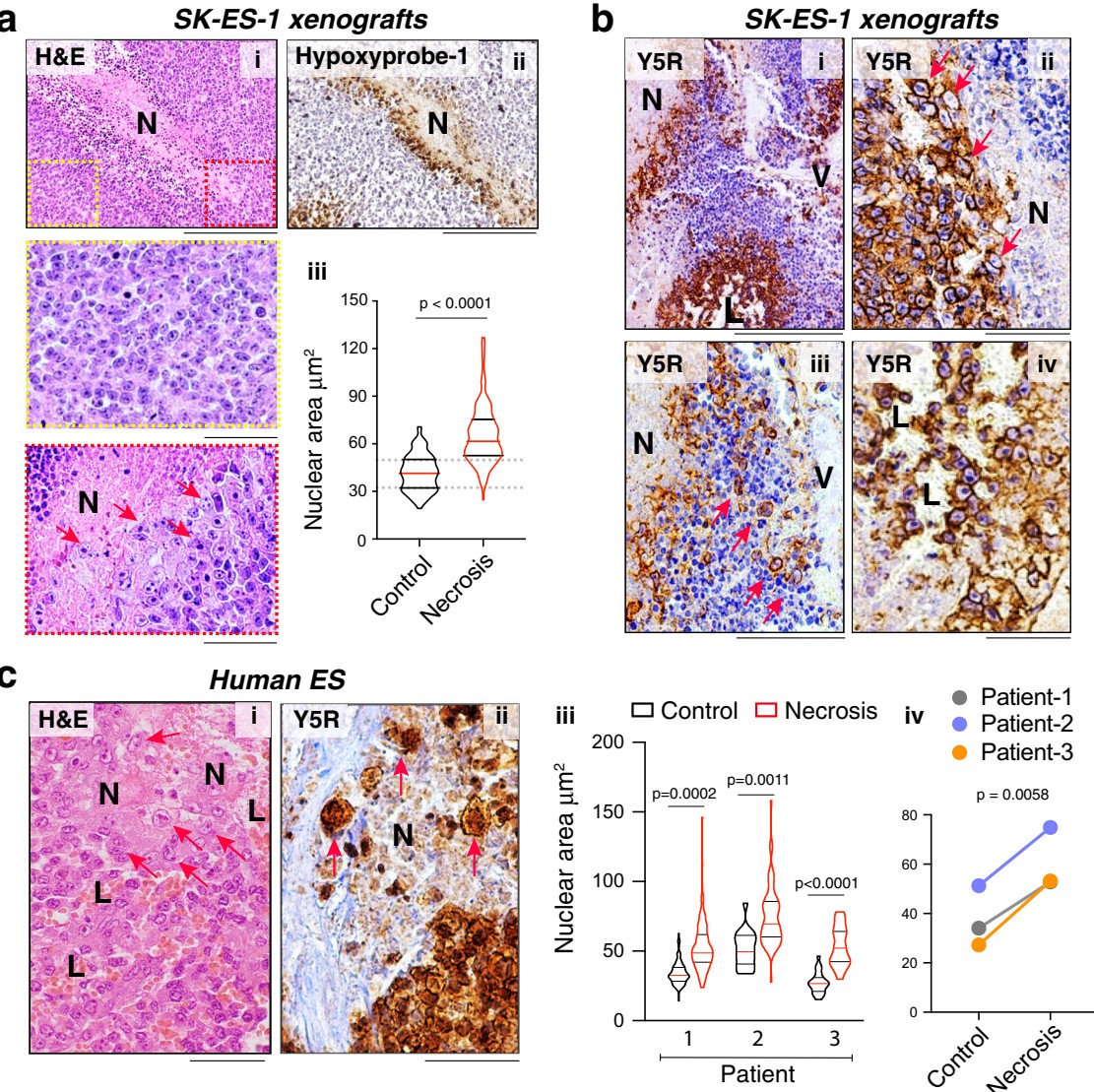

**Fig. 8 Hypertrophic tumor cells accumulate in hypoxic areas of ES tumors. a** Representative images of tumor cells distal to an area of necrosis (Control, yellow inset) and at the border of this region (Necrosis, red inset) in SK-ES-1 xenograft tissue stained with H&E (i) or anti-hypoxyprobe-1 (ii) antibody. Scale bars: 100 μm (i, ii) or 50 μm (insets). Nuclear sizes were compared between tissues in both areas (iii) (Control $n = 101$ and Necrosis $n = 150$; analysis from four independent necrotic areas). Two-sided unpaired $t$-test. **b** Y5R staining in the necrotic area of SK-ES-1 xenografts reveals enlarged, Y5R-positive cells at the edge of this region (i, ii), disseminated throughout it (iii) and accumulated around vessels (i) and blood lakes (i, iv) ($n = 20$ xenografts). Scale bars: 100 μm (i, iii) or 50 μm (ii, iv). **c** Representative image of the edge of necrotic tissue in human ES stained with H&E (i) and immunostained for Y5R (ii). Scale bars: 50 μm. Nuclear sizes in tumor cells distal to a necrotic area (Control) and at the edges of this region (Necrosis) were compared in three human ES tumors not exposed to therapy (Patient 1: Control $n = 23$ and Necrosis $n = 22$; Patient 2: Control $n = 19$ and Necrosis $n = 20$; Patient 3: Control $n = 22$ and Necrosis $n = 18$ cells). The analysis was performed independently within each tissue (iii) and as a group (iv). Two-tailed unpaired t-test. (iii) and two-tailed paired $t$-test (iv). **a–c** N - necrosis, V - blood vessel, L - blood lake; red arrows or borders – hypertrophic cells; yellow borders – tumor cells with normal morphology. For violin plots in **a**, **c**, the red lines represent the median; the black lines represent the quartiles.

xenografts and human tumors, as well as other malignancies. In solid tumors, hypoxic cells are most commonly found on the border of necrotic tissues (Fig. 8ai–ii). Our analysis revealed accumulation of hypertrophic tumor cells in these hypoxic areas of ES xenografts, with the tumor cells on the border of necrosis exhibiting an ~1.5-fold increase in nuclear area (Fig. 8aiii). These cells preserved their proliferative capabilities, as evidenced by expression of the mitogenic marker, Ki67 (Fig. S13a). Similar enlarged Ki67-positive cells were detected in primary tumors exposed to FAL and their corresponding bone, but not soft tissue metastases (Fig. S13b–d) The hypertrophic cells surrounding the necrotic areas were also highly positive for Y5R (Fig. 8bi–ii).

Additionally, the single, Y5R-positive hypertrophic cells were detectable within the necrotic areas (Fig. 8biii) and accumulated around blood vessels and blood lakes (Fig. 8bi, iv), suggesting their migration from the hypoxic environment toward areas with better perfusion.

A similar analysis of human ES samples is challenging, since available tissues are typically limited to pre-treatment biopsies, which contain insufficient material for detailed topographical analysis, and post-treatment surgical specimen that often contain mainly necrotic tissues. To overcome this problem, we focused our analysis on unique large ES specimens derived from tumors not exposed to treatment. In these cases, tumor cells adjacent to

**a** *Neuroblastoma primary tumor - TH-MYCN mice*

**b** *SK-N-BE(2) bone metastasis*

**c** *SK-N-BE(2) cells in vitro*

**Fig. 9 Hypoxia promotes the formation of hypertrophic neuroblastoma cells. a** Enlarged cells in neuroblastoma tissue from the TH-MYCN mouse model stained with H&E or anti-NPY antibody (n = 16 tumors). **b** Bone metastasis tissue from SK-N-BE(2) neuroblastoma xenograft stained with H&E, followed by the analysis of nuclear area of neuroblastoma cells in the main tumor mass (Control) and bone cavity (Bone niche) (n = 45 and 50, respectively). Scale bar: 100 μm. Two-tailed unpaired t-test. **c** Analysis of nuclear area in SK-N-BE(2) neuroblastoma cells cultured in normoxia (NOR) or hypoxia (0.1% oxygen, HYP) for 72 h (n = 100 and 104, respectively). Two-tailed unpaired t-test. **a, b** N - necrosis, V - blood vessel, B - bone, red arrows - hypertrophic cells, yellow arrows - tumor cells with normal morphology. For violin plots in **b, c**, the red lines represent the median; the black lines represent the quartiles.

necrotic areas and accumulating around blood lakes had nuclei enlarged by ~1.5-fold, as compared to the main tumor mass, similar to ES xenografts (Fig. 8c). The same phenomenon was observed in neuroblastoma, another NPY-rich tumor expressing Y5R. Groups of enlarged, NPY-positive neuroblastoma cells were extending between necrotic areas and blood vessels in tumor tissues derived from TH-MYCN mice, which spontaneously develop neuroblastoma (Fig. 9a). Hypertrophic cells were also observed in bone metastasis from SK-N-BE(2) neuroblastoma xenografts (Fig. 9b), while hypoxia in vitro increased the frequency of SK-N-BE(2) cells with enlarged nuclei (Fig. 9c). These data implicate hypoxia-induced polyploidy as a mechanism underlying bone metastasis in various tumors.

## Discussion

Growing evidence indicates the role of tumor hypoxia in ES metastasis[19,21–25]. Hence, we created an in vivo hypoxia model to test the impact of ischemia on ES metastatic pattern and uncover mechanisms underlying these effects[37]. Using this model, we have shown that hypoxia preferentially promotes bone metastasis. These data are in agreement with clinical observations indicating a high prevalence of bone metastases in ES patients with underperfused tumors[19].

Our findings highlight the differences in mechanisms driving metastases to various niches, as previously suggested by the divergence in metastatic patterns of ES xenografts[38]. Recent reports implicated transcriptional changes shifting the balance

away from EWS-FLI1-driven gene expression toward Wnt signaling, as well as the translational activation of pro-metastatic genes induced by Y-box binding protein 1 (YB-1) under cellular stress as metastatic triggers in ES[6–8,25,43]. However, these studies focused on lung metastases and the mechanisms of osseous dissemination remain understudied. Dickkopf-related protein 2 (DKK2) is one of the few factors implicated in ES bone invasiveness, yet its role in distant bone metastasis has not been shown[44]. Furthermore, none of the mechanisms proposed in the aforementioned studies address the role for the acquired CIN in ES metastases. Our data implicate the formation of polyploid cells in the hypoxic tumor environment as a central mechanism promoting ES bone metastasis, since their progeny had an enhanced capability to initiate local bone invasion and distant bone metastases. We were not able to test the metastatic properties of the 2n population from hypoxic ES cells, as they converted to the 4n-like phenotype with passaging. Hence, hypoxia triggers long-term effects, and the subsequent formation of polyploid cells occurs even upon re-oxygenation, as suggested by previous data indicating that tumor cells preserve the cellular memory of hypoxia[45]. Notably, NOR~4n cells did not metastasize, proving that polyploidy alone is not sufficient to trigger metastasis and suggesting that HYP~4n cells have additional features that facilitate this process.

Mechanisms underlying the preferential metastasis of HYP~4n cells to bone remain to be determined. Since low oxygen tension is an important feature of osseous pre-metastatic niche, the high frequency of bone metastases from hypoxic tumors may reflect metabolic adaptation of tumor cells to low oxygen[46–49]. Other mechanisms are likely to include an increased affinity of HYP~4n cells to bone and their ability to invade this tissue. Investigations into these processes may reveal vulnerabilities of the metastatic cells that could be therapeutically targeted[39].

Both HYP~4n tumors and their corresponding metastases exhibited high CIN that mimicked chromosome gains described in aggressive ES[9–11]. Such chromosome changes associate with hypoxia in various malignancies and commonly arise due to mitotic defects that lead to tetraploidy and initiate a characteristic genome evolution, as divisions of these tetraploid intermediates result in abnormal chromosome segregation leading to their losses and CIN[13,14,50]. With time, this process results in monosomies that hinder cell growth and trigger a subsequent round of polyploidization. These changes confer an advantage to tumor cells and allow them to overcome selective pressures, such as hypoxia or cytotoxic therapy[51,52]. In line with this, our data indicate that polyploid cells survive hypoxia in ES xenografts and that their progeny initiate bone metastases. Notably, with prolonged passaging in vitro, HYP~4n cells lost their metastatic properties, which is in line with acquiring unfavorable monosomies and a need for the next polyploidization to regain fitness[14]. Hence, the enhanced metastatic potential of HYP~4n cells is a transient state. This phenomenon may explain why bone-derived TC71 cells required another hypoxic stimulus to trigger osseous metastasis, despite their near-triploid karyotype[53]. In addition to their metastatic properties, HYP~4n cells were chemoresistant, as shown in other tumor types[51,52]. Thus, hypoxia-induced changes may contribute to the disease recurrence and secondary ES dissemination, providing a rationale for targeting this cell population.

We have previously established that despite a high expression of NPY and Y5R in ES cells, selective activation of this pathway requires hypoxia[23,54]. Under these conditions, NPY stimulated the migration of ES cells with a cancer stem cell phenotype via the Y2R/Y5R axis[23]. We have now shown that in addition to this Y2/Y5R-induced cell motility, hypoxia can trigger selective stimulation of Y5R, which over-activates the RhoA pathway, leading to

cytokinesis defects and polyploidization. As a central cytoskeleton regulator, RhoA is tightly controlled during cytokinesis. Initially, high RhoA activity in the cleavage furrow drives its ingression, with subsequent inhibition of the RhoA being crucial for detachment of the daughter cells[42]. Our data indicated a defect in this last stage of cytokinesis in CHO-K1/Y5R transfectants, leading to the formation of multinucleated and polyploid cells. A similar phenotype was observed in hypoxic ES, in association with a Y5R-dependent increase in RhoA activity. In addition to inducing polyploidy, RhoA activation may overcome the p53-mediated tetraploidy checkpoint by inhibiting its negative regulator, LATS2[55]. The survival of the polyploid cells and their subsequent genome evolution may also be facilitated by TP53 and STAG2 gene mutations, respectively, both of which associate with worse prognosis in ES patients[5,18]. While the metastatic effect of STAG2 loss has been recently associated with transcriptional reprogramming, previous reports indicated increased copy number aberrations in ES tumors with TP53 and STAG2 mutations[18,56]. Moreover, the impact of STAG2 loss on chromosomal segregation in polyploid cells has never been addressed.

The role of the RhoA pathway in ES progression has been implicated by previous studies. EWS-FLI1 represses Rho-actin signaling, while metastatic cells with low EWS-FLI1 have an increased activity of this pathway[8]. As a LATS2 inhibitor, RhoA can also activate downstream targets repressed by this factor, i.e. YAP/TAZ transcription regulators, which promote ES invasiveness[55,57]. It remains to be determined whether the hypoxia-driven NPY/Y5R signaling provides an alternative mechanism of RhoA activation in the presence of high EWS-FLI1, or if it is associated with its down-regulation, as shown for lung metastases[6–8]. Earlier reports indicating an increase in EWS-FLI1 expression in hypoxic ES cells favor the former hypothesis[21].

The role of the NPY/Y5R pathway in ES osseous dissemination is supported by our previous clinical and preclinical data. NPY serum levels were higher in ES patients with pelvic and axial tumors originating in bone, as compared to those with extra-osseous lesions[58]. Likewise, expression of the NPY system was elevated in tissues from bone ES. In an ES orthotopic model, high endogenous NPY expression resulted in increased bone invasion and osseous dissemination, while NPY knock-down reduced the bone degradation induced by the tumor[38]. The current data provide a mechanism for the role of the NPY system in osseous metastasis. However, its direct impact on bone remodeling in the context of ES needs to be investigated. Importantly, NPY regulates bone homeostasis via direct effects on osteoblasts and its central activity in the brain[59].

In the current study, a Y5R antagonist was administered for only five days, during the time when xenografts develop endogenous hypoxia, yet it successfully reduced hypoxia-induced polyploidization and prevented dissemination to bone[37]. Moreover, the dose of the antagonist had to be limited due to its anti-orexigenic activity[60]. It is therefore possible that a longer and more effective Y5R inhibition (e.g. using antagonists that do not cross blood brain barrier) may have an effect on the overall metastatic potential of ES cells, particularly given the inhibitory effect of the Y5R antagonist on ES cancer stem cell motility[23]. Notably, Y5R has been implicated in chemoresistance[26]. Hence, our data warrants further investigation into its role in ES dissemination and recurrence.

In addition to ES, the expression of NPY and its Y5R has been shown in other cancers. In neuroblastoma, high NPY release associates with metastases and relapse, while Y5R expression is elevated in chemoresistant tumors and cells with an angioinvasive phenotype[26,61,62]. Here, we have shown that like ES, neuroblastoma tumors contain hypertrophic cells, which arise in hypoxia. In breast and hepatic cancers, Y5R promotes cell

**Fig. 10 Proposed mechanisms leading to hypoxia-induced bone metastasis in ES.** Hypoxic exposure up regulates the expression of NPY and Y5R in ES cells, leading to over-activation of the NPY/Y5R/RhoA axis and cytokinesis failure. The resulting tetraploid cells undergo abnormal cell divisions, which lead to chromosomal instability. This process may be facilitated by defects in the p53 pathway and STAG2 mutation, which enables tetraploid cell survival and subsequent chromosome loss, respectively. The progeny of hypoxia-induced polyploid cells have an increased ability to survive in low oxygen, invade bone tissue and form distant osseous metastases.

proliferation and migration[29,33,63,64]. High systemic levels of NPY are also associated with advanced stage prostate cancer[31,65,66]. Notably, patients with the above cancers frequently present with treatment-resistant osseous metastases, suggesting that the mechanisms uncovered in ES may be relevant to other malignancies.

In summary, we have provided evidence for the role of hypoxia in promoting ES bone metastases and identified a specific mechanism leading to their development, which is different from those responsible for extra-osseous dissemination. We have shown that tumor hypoxia stimulates the NPY/Y5R axis, which leads to RhoA over-activation, cytokinesis defects and polyploidy (Fig. 10). The progeny of these hypoxia-induced polyploid cells have high CIN and a capacity to colonize the bone environment. Further studies are required to identify specific vulnerabilities of this cell population, which then can be exploited in therapies targeting existing bone metastases. Our findings provide the direct link between known adverse prognostic factors in ES, namely metastases, CIN and tumor hypoxia, and implicate Y5R as a potential target for treatments preventing the hypoxia-induced evolution of the cancer genome and subsequent disease progression. The mechanisms described herein may be relevant to other malignancies known to form bone metastases, which remain intractable and significantly impact patient prognosis quality of life.

## Methods
**Materials**. NPY was purchased from Bachem (San Carlos, CA). Y5R agonist, BWX 46, Y5R antagonist, CGP71683, and Y2R agonist, NPY$_{13-36}$, were obtained from Tocris (Ellisville, MO). Y1R agonist, [Arg6, Pro34]NPY, was provided by Dr. Annette Beck-Sickinger, University of Leipzig, Germany. Rho inhibitor I, C3 transferase, was purchased from Cytoskeleton, Inc. (Denver, CO).

**Cell culture**. Human ES cell line, SK-N-MC (HTB-10), human neuroblastoma cell line, SK-N-BE(2) (CRL-2271), and Chinese hamster ovary cells, CHO-K1 (CCL-61), were obtained from American Type Culture Collection (ATCC, Manassas, VA) and cultured according to the supplier's recommendation. Other ES cell lines were obtained from Dr. Jeffrey Toretsky (Georgetown University Medical Center). SK-ES-1 cells were cultured in McCoy's 5A modified medium supplemented with 15% fetal bovine serum (FBS) in collagen-coated flasks, while all other ES cells were maintained in RPMI medium with 10% FBS[67]. All media were supplemented with penicillin (200 units/mL), streptomycin (200 µg/mL), and fungizone (1 µg/mL). Primary cultures of ES cell lines were developed from tissue fragments of ES xenografts – primary tumors and metastases - in their corresponding culture media[37]. Hypoxia was created in an O$_2$ Control InVitro Cabinet (COY Lab Products, Grass Lake, MI) equipped in oxygen sensor regulating a flow of 95% N$_2$ and 5% CO$_2$ gas mixture to maintain 0.1% O$_2$ levels through the duration of the experiments.

**CRISPR/Cas9 NPY5R gene editing**. SK-ES-1 cells were transduced with lentiviral vectors encoding doxycycline (Dox)-inducible Cas9 nuclease (hEF1α-Blast-Cas9 Nuclease) and single-guide RNA (sgRNA) against human NPY5R gene (GSGH11838-Edit-R, clone # VSGHSM_26628100, 5′-ACTACGG-TAAACTTCCTCAT-3′; Dharmacon Horizon Discovery, Lafayette, CO). The specificity of the selected Y5R sgRNA was confirmed using the Cas-OFFinder tool[68]. The stably transfected cells (SK-ES-1/Dox-Cas9/Y5R sgRNA) were selected based on blasticidin (5ug/ml) and puromycin (1ug/ml) resistance. Then, the heterogenous pool of SK-ES-1/Dox-Cas9/Y5R sgRNA cells with edited NPY5R gene was obtained by Dox treatment (1µg/ml, 96h). The profile of Cas9-targeted NPY5R gene disruption in the tumor cell population was evaluated with next-generation amplicon sequencing using the Illumina MiniSeq System. Gene editing efficiency in the heterogeneous population of tumor cells was analyzed with Crispresso2 software[69].

**Orthotopic ES xenograft model**. All procedures involved in animal experiments were approved by the Georgetown University Institutional Animal Care and Use Committee. Orthotopic ES xenograft and FAL were performed as previously described[37,38]. In all, $2 \times 10^6$ (SK-ES-1 and TC32) or $1 \times 10^6$ (TC71) of ES cells suspended in 0.1 ml of saline were injected into gastrocnemius muscles of 3-4 weeks old female SCID/beige mice (Charles River Laboratories, Wilmington, MA), adjacent to the left proximal tibial crest. The growth of primary tumors was monitored by periodical measurements of the limb and the volume of tumors was calculated according to the formula $(D \times d^2/6) \times \pi$, where $D$ was the longer diameter and $d$ was the shorter diameter. Once tumors reached a volume of 150 mm$^3$ (SK-ES-1) or 250 mm$^3$ (TC71), tumor-bearing limbs from control mice were amputated. To assess hypoxia, 3 h before amputation, the mice were injected intraperitoneally with hypoxyprobe-1 (pimonidazole; HPI, Burlington, MA), 1.5 mg/mouse. In the hypoxia groups, mice were subjected to FAL proximal to the bifurcation of the profunda femoris and femoral artery and the amputation was performed 3 days later. Metastasis formation was monitored by periodic MRI for 30–50 days, depending on the cell line. Mice were euthanized at the end of the experiment or when significant metastasis burden was detected. Macroscopic metastases, organs and bones were harvested and fixed in 10% buffered formalin for histopathological analyses. To test the effect of Y5R inhibition on ES metastases, once primary tumors reached a volume of 150 mm$^3$, mice were randomized into two groups and treated with vehicle or Y5R antagonist, CGP71683, 20 mg/kg, administered for 5 days by daily intraperitoneal injections, followed by limb amputation and subsequent metastasis monitoring. Alternatively, mice were injected with SK-ES-1 or SK-ES-1/Dox-Cas9/Y5R sgRNA cells and fed with the control diet or pellets supplemented with 625 mg/kg doxycycline (Envigo, Frederick, MD), starting 7 days before injection. Once tumors reached a volume of 150 mm$^3$, mice on either the Dox- or Dox+ diet were randomized into control and FAL-treated groups. Metastases were monitored as above.

**Neuroblastoma animal models**. 129×1/SvJ mice expressing the human MYCN oncogene under a rat tyrosine hydroxylase promoter (TH-MYCN mice) were obtained from the National Cancer Institute (Frederick, MD)[70]. Both male and female mice at the age of 6–12 weeks were used for tumor tissue analyses. In the orthotopic neuroblastoma xenograft model, $2 \times 10^5$ of SK-N-BE(2) cells suspended in 10 µl of Matrigel (Corning Inc., Corning, NY) were injected into adrenal fat pad of 3–4-weeks-old female SCID/beige mice (Charles River Laboratories)[71]. In both models, tumor growth and metastasis were monitored by MRI, and the animals were euthanized once primary tumors reached a volume of 1000 mm$^3$. Tissues from primary tumors and metastases were fixed and subjected to histopathological analyses.

**MRI**. MRI was performed in the Georgetown University Lombardi Cancer Center Preclinical Imaging Research Laboratory on a 7 Tesla Bruker horizontal spectrometer run by Paravision 5.1 software, as previously described[72–76]. During imaging mice were anesthetized with 2% isoflurane, 30% oxygen, and 70% nitrous oxide, placed on a holder with respiration monitoring and imaged either in a 40 or 23 mm Bruker mouse volume coil for whole body and brain imaging. The anatomical imaging protocols used were two-dimensional, T2-weighted RARE sequences that included the following parameters: TR = 3000 ms, TE = 24 ms, matrix = 256 × 256, FOV = 4.35 × 3.0 cm, slice thickness = 0.5 mm, RARE factor = 4, and averages = 4.

**Human ES tissue samples. Human ES tissue samples**. The human ES samples were obtained by Dr. Iżycka-Świeszewska (Medical University of Gdańsk, Poland) in compliance with institutional ethical regulations provided by Medical University of Gdańsk Bioethics Committee, which approved their use for research (protocol# NKBBN/448/2015). Use of these samples was further reviewed and approved by Georgetown University Institutional Review Board. The study cohort contained archival formalin-fixed paraffin-embedded (FFPE) tissue sections from human ES, which were collected for clinical diagnostic purposes in years 2002–2011 and fully de-identified. Hence, according to the definition provided by the U.S. Department of Health & Human Services, this study was not considered human subjects research and no informed consent or a waiver was required. 19 ES cases with confirmed t(11,22) chromosomal translocation corresponding to the EWS-FLI1 gene fusion were assessed and three surgical specimens excised from tumors not subjected to treatment were selected for further analyses.

**Tissue analyses**. For the metastasis detection in animal models, all organs, long bones and vertebral columns, as well as other macroscopic bone and soft tissue metastases detected by MRI or during necropsy, were collected, sectioned, and stained with H&E. The bones were longitudinally cut to expose bone marrow cavity at the entire bone length. The presence of metastases was assessed based on H&E staining. The metastases were classified as osseous when MRI at early stages of their development and/or histopathological analysis indicated their bone origin, in contrast to the secondary lesions that originated in soft tissues and exhibited local bone invasion at their late stages. Immunostaining was performed on FFPE tissues using mouse monoclonal anti-CD99 (Clone 12 E7, IR057, DAKO, Carpinteria, CA; 1:50) and anti-CD68 (Clone KP1, GA609, DAKO, ready to use) antibody, as well as rabbit polyclonal anti-pimonidazole (Hypoxyprobe™-1 kit; HPI, Inc. Burlington, MA; 1:300), anti-CAIX (ab15086, Abcam, Cambridge, MA; 1:1000), anti-Y5R (NB1-00957, Novus Biologicals, Littleton, CO; 1:300) and anti-Ki67 (ab15580, Abcam; 1:100) antibody. Tissue sections from xenografts and human ES were assessed by two independent pathologists.

The histopathological analysis of the tumor tissue surrounding necrotic areas and blood lakes known to develop due to prior tumor hypoxia focused on hypertrophic, yet viable cells. The cell morphology was considered as consistent with polyploidy based on an enlarged nucleus (>1.5 fold of average nuclear area), clear hyperchromatic nucleus with the regular chromatin pattern, often visible nucleolus or multiple nucleoli with clear borders, and narrow bright cytoplasm with an intact nuclear membrane. The cells with pyknotic nuclei were not taken into consideration. Moreover, immunostaining with anti-CD99 and anti-CD68 antibody was performed to confirm the tumoral origin of these cells and exclude the macrophages, respectively. Furthermore, Ki67 staining was performed to detect actively proliferating cells.

**Nuclear size and mitotic segregation error analysis**. Cells were cultured on glass cover slips, stained with wheat germ agglutinin (WGA; Thermo Fisher Scientific, Waltham, MA) at a concentration of 1.2 µg/ml and then fixed in 4% paraformaldehyde and stained with DAPI (Thermo Fisher Scientific) at a concentration of 2 µg/ml. Nuclear sizes were measured using ImageJ software (NIH, Bethesda, MD) based on DAPI staining, while WGA staining allowed for identification of multinucleated cells. Parallel analysis was performed on H&E stained tumor tissues. The nuclear size of the cells in the main tumor mass, in bone invasion areas and on the borders of necrosis were measured using ImageJ software. Mitotic segregation errors and abnormalities in the morphology of interphase nuclei were assessed based on DAPI staining under Zeiss Axioskop fluorescence microscope (Zeiss, White Plains, NY).

**DNA content**. Cells were fixed in 75% ethanol, and stained with propidium iodide (PI) according to the standard procedures. Flow cytometric analysis was done on LSR-Fortessa (Becton Dickinson, Franklin Lakes, NJ) and data analyzed using FCS Express and ModFit LT software packages (Verity Software House Inc., Topsham, ME). Gating strategy is depicted in Fig. S14.

**FISH analysis**. Fluorescence in situ hybridization (FISH) assays were performed to evaluate gene copy number and chromosome ploidy in the primary and metastatic FFPE tissue samples and in the ES cell lines. DeepSee Quad probe (Enzo Life Sciences, Farmingdale, NY), containing probes for the CDKN2A locus at 9p21 and centromeres of chromosomes 3, 7 and 17 (CDKN2A/CEN3/7/17), were hybridized to harvested interphase cells from cell lines fixed in methanol-acetic acid fixative and 5µm FFPE tumor sections using a laboratory standardized protocol with slight modifications[77]. At least 100 intact nuclei per case were captured and evaluated using the BioView Duet automated imaging system (BioView, Billerica, MA).

For FISH analysis of the NPY5R gene (mapped at 4q32.2) in ES cells, a FISH probe was constructed, using a BAC clone (RP11-638L1) containing sequences of the NPY5R gene (BACPAC Resources, Oakland, CA). For the centromere 4 probe (Chr4p11-p12, CEP 613 kb), a directly labeled 'SureFISH' orange red probe was hybridized according to the manufacturer's protocol (Agilent Technologies, Santa Clara, CA). BAC clone DNA was prepared and labeled with biotin-16-dUTP (Roche Applied Sciences, Indianapolis, IN) using nick translation[78]. Probes were hybridized to interphase ES cells· The biotin-labeled probe was visualized with avidin conjugated to FITC (Vector Laboratories, Burlingame, CA) and nuclei were counterstained with DAPI. Digital image acquisition was performed using a 100X objective mounted on a Leica DMRBE microscope (Leica, Wetzlar, Germany) equipped with optical filters for DAPI Fluorescein, FITC, and TRITC (Chroma Technologies, Brattleboro, VT) and a cooled charge-coupled device (CCD) camera (Photometrics, Tucson, AZ). The IPLab software package (Scanalytics Inc, Fairfax, VA) was used for image acquisition and processing. Slides were scored by two independent observers. A minimum of 50 nuclei were evaluated in each case. Only intact non-overlapping nuclei were scored. The detection of two FISH signals was considered as normal copy number, and three or more signals as gain or amplification.

**Glucose measurement**. A volume of 20 µL of the media was mixed with 100 µL of MeOH containing the internal standard (4-Nitrobenzoic acid) which was prepared in MeOH at a concentration of 30 µg/mL and homogenized for 60 s. The samples were transferred to the GC vials then dried under vacuum at 35 °C. Dry samples were derivatized by adding 10 µL of methoxyamine (20 mg/mL) then heated in an agitator at 60 °C for 30 min. This was followed by 100 µL of MSTFA. The vials were transferred to an agitator to heat at 60 °C for 30 more minutes. Finally, the vials were capped and a volume of 2 µL was injected directly to the GCMS in a splitless mode into an Agilent 7890B GC system (Santa Clara, CA, USA) that was coupled with a Pegasus HT TOF-MS (LECO Corporation, St. Joseph, MI, USA). Separation was achieved on a Rtx-5 w/Integra-Guard capillary column (30 m × 0.25 mm ID, 0.25 µm film thickness: Restek Corporation, Bellefonte, PA, USA), with helium as the carrier gas at a constant flow rate of 0.8 mL/min. The temperature of injection, transfer interface, and ion source was set to 150, 270, and 320 °C, respectively. The GC temperature programming was set to 0.2 min. of isothermal heating at 70 °C, followed by 6 °C/min oven temperature ramping to 300 °C, a 1.0 min. isothermal heating of 300 °C, 20 °C/min to 320 °C, and a 2.0 min. isothermal heating of 320 °C. Electron impact ionization (70 eV) at full scan mode (40–600 m/z) was used, with an acquisition rate of 20 spectra per second in the TOF/MS setting. The collected spectra were processed, aligned, and deconvoluted with the software ChromaTOF v. 4.72.0.

**Chromosome count**. CHO-K1 cells were grown to 80% confluency and incubated with colcemid at a concentration of 1 µg/ml for 2 h and the metaphase chromosome spreads were prepared according to the standard procedures[78,79]. Chromosome counts from at least 3 independently developed clones were compared between the experimental groups.

**Isolation of diploid and tetraploid cell fractions**. ES cells were transduced with Premo™ FUCCI Cell Cycle Sensor BacMam 2.0 vectors encoding geminin-GFP and Cdt1-RFP (Thermo Fisher Scientific), according to the manufacturer's protocol, and incubated overnight. Transduced cells were subsequently incubated in normoxia or hypoxia for 24 h and stained with Hoechst 33342 (Thermo Fisher Scientific) at a concentration of 2 µg/ml. Cells were then sorted into 2c (2n) and 4c/GFP-/RFP + (4n) populations using FACSAria (Becton Dickinson).

**Immunocytochemistry**. Cells were cultured on glass cover slips, fixed in 4% (w/v) paraformaldehyde and stained with rabbit polyclonal anti-Y5R (ab133757, Abcam; 1:250) and mouse monoclonal anti-RhoA-GTP (NewEast Biosciences King of Prussia, PA; 1:100) antibodies. DNA was stained using DAPI at 0.5 µg/ml in PBS.

**Western blot**. Western blot for Y5R was performed on membrane proteins, isolated as previously described[80]. To this end, cells were lysed in the buffer containing 10 mM Tris-HCl, pH 7.6, 5 mM EDTA, 3 mM EGTA, 250 mM sucrose and the protease inhibitor cocktail (Sigma, St. Louis, MO). The lysate was incubated for 15 min on ice and sonicated, followed by centrifugation at 500 g for 5 min at 4 °C. The supernatant was then collected and centrifuged at 20,000 × g for 30 min at 4 °C. The cell membrane pellet was incubated in the buffer containing 20 mM HEPES, pH 7.4, 150 mM NaCl, 5 mM EDTA, 3 mM EGTA, 4 mg/ml β-dodecylmaltoside and protease inhibitors for 1 h on ice and centrifuged at 20,000 × g for 30 min at 4 °C. The resulting membrane protein solution was subsequently used for Western blot with goat polyclonal anti-Y5R antibody (EB06769; Everest Biotech, Ramona, CA; 1:1000), while mouse monoclonal antibody against alpha 1 sodium potassium ATPase (ab7671, Abcam; 1:1000) served as a loading control. Alternatively, whole cell extracts were used with a mouse monoclonal anti-β-actin antibody (A1978, Sigma; 1:10000) as a control. Densitometry was conducted using ImageJ software.

**Transfections**. SK-ES-1 cells were transfected with pEGFP-N1 or pmCherry-N1 vectors (Takara, San Jose, CA) using Lipofectamine 2000 (Invitrogen, Carlsbad,

CA). Stable transfectants were selected with geneticin (Sigma, St. Louis, MO) at a concentration of 0.25 mg/ml. Transfection of CHO-K1 cells with NPY receptors was performed as previously described[41]. cDNAs of human Y1R, Y2R and Y5R were cloned into pEGFP-N1 vector at NheI and BamHI restriction sites. Transfection was performed using Lipofectamine 2000 (Invitrogen). Stable transfectants carrying EGFP-fused NPY receptors were selected with geneticin (Sigma) at a concentration of 1 mg/ml followed by single cell clone isolation. The membrane localization of the Y5R-EGFP fusion protein in the selected clones was confirmed by co-staining with WGA (1.2 µg/ml, 5 min), followed by fixation with 4% paraformaldehyde and confocal microscopy using Olympus Fluoview BX61 microscope (Olympus, Center Valley, PA).

**Time lapse microscopy**. CHO-K1 cells were transfected with vectors encoding EGFP fused to NPY receptors. 4 h later, several fields with EGFP-positive cells were selected and imaged with Nikon TE300 microscope (Nikon Instruments Inc., Melville, NY) for 96 h. Imaging was performed using both bright field and fluorescence settings to identify EGFP-positive cells. Data was acquired using Metamorph software (Molecular Devices, San Jose, CA).

**p44/42 signaling**. Stable Y5R transfectants were treated with NPY at a concentration of $10^{-7}$M for the desired time (1-90 min). Cells were lysed immediately after treatment. Total and phosphorylated p44/42 MAPK were detected in cell lysates using mouse monoclonal anti-phospho p44/42 E10 antibody (9106, Cell Signaling Technologies, Danvers, MA; 1:2000) and rabbit polyclonal anti-p44/42 MAPK antibodies (9102, Cell Signaling Technologies; 1:1000). The band intensities were quantified by densitometry using ImageJ software.

**Proliferation and viability assays**. CHO-K1 cells stably transfected with Y5R cDNA were plated in 96-well plates, cultured in 1% FBS media for 24 h and treated with a range of NPY concentrations. After 24 h, the number of viable cells was assessed using MTS-based CellTiter 96®AQueous One Solution Cell Proliferation Assay (Promega, Madison, WI). Additionally, $^3$H-thymidine uptake under the same conditions was measured. To this end, 0.5 µCi of [$^3$H]thymidine per well was added for the last 6 h of incubation and the cells were harvested in a 96-well harvester (Tomtec, Hamden, CT) and counted in a Wallac 1205 Betaplate Liquid Scintillation Counter (Perkin Elmer, Boston, MA)[81–83]. The effect of NPY ($10^{-7}$M) or Y5R antagonist ($10^{-6}$M) on cell number was determined by cell count 48 h after culture in various serum concentrations (0, 1, 10% FBS). To assess DNA synthesis in situ, CHO-K1/Y5R transfectants were cultured on glass cover slips, incubated with 10 µM EdU (Thermo Fisher Scientific) for 1 h and stained according to the manufacturer's protocol. To assess resistance to chemotherapy, ES cells were plated in 96-well or 24-well plates in the standard growth medium, treated with a range of doxorubicin concentrations (0.05–0.4 µg/ml) and cultured for 72 h. The number of viable cells was assessed by MTS assay or cell count upon trypan blue exclusion.

**RhoA pull-down assay**. CHO-K1 cells, wild type or stably transfected with Y5R-EGFP, were incubated for 24 h in serum-free media, followed by stimulation with $10^{-7}$M NPY for 20 min or $10^{-6}$M Y5R antagonist, CGP71683, for 30 min. Subsequently, cells were lysed and levels of RhoA-GTP were assessed using RhoA Pull-down Activation Assay Biochem Kit (Cytoskeleton Inc.) according to the manufacturer's protocol. SK-ES-1 cells were either pre-incubated in serum-free media for 24 h and then treated with selective NPY receptor agonists at a concentration of $10^{-7}$M for 20 min, or cultured in media containing 10% FBS and incubated in normoxia or hypoxia for the desired time, in the presence or absence of $10^{-6}$M Y5R antagonist or Rho inhibitor I (0.01 µg/ml). Subsequently, cells were lysed and the levels of active RhoA were assessed as above.

**Statistical analyses**. Statistical analyses were performed using SAS 9.4 (SAS Inc, NC) and GraphPad Prism 6 software. Between-group comparisons were assessed using one-way repeated measures ANOVA with post-hoc t-test, independent-samples t-tests or paired-samples t-tests. For count data, Fisher's exact, $\chi^2$ or Mann–Whitney tests were used, as appropriate. The generalized estimating equation (GEE) models were used to compare the treatment effect and time trend on the number of metastases (total, bone, extra-osseous) developed over time between control and treatment groups. Significant associations were assessed at an alpha level of 0.05. All in vitro experiments were repeated at least three times. Animal experiments were performed twice and the combined results from both independent experiments are shown. Data is presented as mean ± standard errors.

**Reporting summary**. Further information on research design is available in the Nature Research Reporting Summary linked to this article.

## Data availability
The raw images of tissue and cell staining, as well as MRI data are available from the author upon request. All other data are available in the article, Supplementary Information or Source data file. Source data are provided with this paper.

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

## Acknowledgements

This work was supported by NIH grants: 1RO1CA123211, 1R03CA178809, R01CA197964, and 1R21CA198698, as well as grants from Sunbeam Foundation and Children's Cancer Foundation to JK. The study was performed in collaboration with the Georgetown-Lombardi Comprehensive Cancer Center's Preclinical Imaging Research Laboratory (PIRL), the Histopathology & Tissue Shared Resource (HTSR), the Flow Cytometry & Cell Sorting Shared Resource (FCSR), the Tissue Culture and Biobanking Shared Resource (TCBSR), Mass Spectrometry and Analytical Pharmacology Shared Resource (MSAP) and the Microscopy & Imaging Shared Resource (MISR), all supported by NIH/NCI grant P30-CA051008. The authors would also like to thank Dr. Yichien Lee for technical help with MRI, as well as Dr. Karen Creswell and Dan Xun for assistance with flow cytometry.

## Author contributions

C.L., A.M., S.H.H., L.R.C., S.D.P., C.A., O.R., M.R., M.G., A.C., and J.K. designed experiments. C.L., A.M., S.H.H., S.G., S.Z., J.U.T., N.A., M.A., S.C., N.E., J.R., M.R., V.Z., J.B., E.K., G.I.G., and M.G. performed experiments. C.L., A.M., S.H.H., S.G., O.R., M.R., E.I.S., S.D.P., and J.K. analyzed data. H.W. performed statistical analyses. C.L., A.M., S.H.H., S.G., J.U.T., O.R., C.A., E.I.S., L.C., S.D.P., and J.K. prepared figures and wrote the manuscript.

## Competing interests

The authors declare no competing interests.

## Additional information

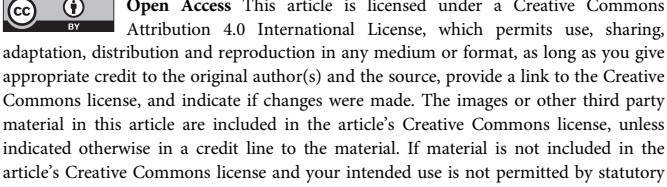

