## [Peer Review File · Nature Communications]

REVIEWER COMMENTS

Reviewer #1 (Remarks to the Author):

The manuscript from Lu et al. describes a study in which the role of NPY receptors in hypoxia-mediated metastasis in Ewing's Sarcoma (ES) is investigated. The authors show data that link hypoxia to aneuploidy and CIN, which is mechanistically linked to NPY receptor activity in in vitro studies, before confirming a role for NPY receptors in metastasis in their in vivo model of ES. This study builds on previous data showing hypoxia-driven genomic instability, including in particular a seminal study from the Bristow research group (Nature Genetics 2019) in which this is described in multiple tumour types. As the authors point out, the effects of hypoxia in ES both in vitro and in vivo are relatively well described, although specific links between hypoxia and CIN in this cancer are not yet characterised. The highly novel aspect of this study is the link between hypoxia, NPY signalling, aneuploidy and metastasis. Overall, this is an interesting study, much of which has been carried out to a very high standard. However, there are some gaps in the evidence that in my view need to be addressed in order for the proposed mechanism to be entirely convincing.

A very general point is that how hypoxia in vitro was elicited in vitro is inadequately described, with the description of a 'chamber' at 0.1% O₂ (in materials and methods) an insufficient level of detail to understand whether this system was artefact-free. Studies of the effects of hypoxia on any aspect of cell phenotype where oxidative stress could play a role require particularly close attention paid to the system in use. As the equipment isn't detailed, we cannot assess whether 0.1% O₂ is an accurate, and more importantly a maintained level of oxygen tension over the timecourse of the experiment (72h of 0.1% O₂ as described for Figure 2D-F). Given that cells in hypoxia use oxygen, the oxygen tension would need to be actively monitored and 'topped-up' to avoid anoxia. Over 72h of severe hypoxia, significant amounts of other nutrients are used from the medium – most importantly perhaps the glucose, which could well be exhausted within 48h or less. This is known to increase CIN, and therefore measurements of glucose (at least) in the medium at the end of the experiment would be important to demonstrate that hypoxia alone resulted in the phenotypes observed – given that the in vivo model was also hyponutrientic (i.e. ischemic).

Figure 1G – an average number of around 0.3-0.4 bone metastases/animal is shown, while in Figure 1H around 30% of animals were suggested to have bone metastases. This suggests that around 6 animals had a single detectable metastasis, given an n=19. It isn't clear from the materials and methods how thoroughly the bones were checked for metastases, and the results could therefore show a higher rate of metastatic growth within the bone, rather than the metastatic spread induced by aneuploidy. This is an important point as the results may therefore reflect cell proliferation in the metastases rather than the process of metastasis.

In Supplementary Figure 4, the authors show changes in cell viability in normoxia and hypoxia after doxorubicin treatment. Firstly, cross-comparison of normoxia and hypoxia is required in order to appreciate how hypoxia alters therapy response, and how the ploidy affected this. Secondly, as MTS is based on cell redox, the signal for the same cell in hypoxia will be altered, so is not a suitable assay to use for hypoxia studies.

The CHO model used to isolate the effect of Y5R expression on rate of aneuploidy also requires some clarification. It isn't clear that CHO cells express NPY for autocrine/paracrine activation of the exogenously-expressed receptor. If not, how are the authors proposing that the receptor is activated in

this system? The microscopy images shown are not of sufficient resolution to confirm cell surface expression of Y5R – it is clearly key that the receptor is appropriately sited in this system. Indeed, it appears that expression is often localised to other intracellular structures (Figure 5A), and perhaps even the mitotic machinery.

In Supplementary Figure 6, where the effect of the Rho inhibitor on cellular DNA content was observed, the traces must be overlaid so that a comparison can be made by the reader. As it stands, the non-quantified axis does not allow the comparison. As another comment, is peak shift the most appropriate measure to use? As it stands, the conclusion might be that the Rho inhibitor reduces the 4c peak to more like a 3.8 peak (although it isn't possible to tell what the result of the Rho inhibitor is with the axes presented) – this isn't necessarily the outcome required, given that 3.8 is still substantially more than 2c. In Figure 7, an elegant system in which Dox-inducible CRISPR-mediated Y5R mutation is utilised in vivo to confirm the mechanism by which hypoxia leads to aneuploidy and metastasis. Figure 7F requires a statistical comparison between Dox- and Dox+ in order to confirm that loss of Y5R alters rate of bone metastasis.

Reviewer #2 (Remarks to the Author):

Review of Lu et al, "Hypoxia-activated neuropeptide Y/Y5 receptor/RhoA pathway triggers chromosomal instability and bone metastasis in Ewing sarcoma". In this manuscript, the investigators examine the links between tumor hypoxia, chromosomal instability and osseous metastasis in Ewing sarcoma, building on the group's previous work examining the role of NPY signaling in the disease. They use a variety of in vitro and in vivo approaches to dissect the contribution of hypoxia and to establish a role for the Y5 neuropeptide receptor and RhoA signaling in altering cell division properties, leading to cells with >4c DNA content and a propensity for bony metastasis. A particularly strong piece of data is the ability of in vivo CRISPR editing of Y5R to prevent osseous metastasis in xenograft models. Overall the study is very well-done and represents an important advance for the field. There are some areas of ambiguity that should be addressed to increase the impact of the manuscript.

Specific comments:

1. How was 3 days of time between FAL and leg amputation arrived at as an experimental timepoint?
2. If large tumors exhibit high endogenous hypoxia, do they give rise to more osseous metastasis?
3. For the analyses in Figure 2, please provide more detail on the experimental and statistical design. How many nuclei were measured for the analyses in Figures 2A and 2D? (Consider providing the data points as part of the violin plots). Please provide n of the number of cells scored for bar charts in Figures 2B, 2C, 2E (methods section for FISH mentions at least 100 nuclei scored, was this also the case for scoring >4c DNA content, or was that based on integration of FACS plot?)
4. In the in vitro hypoxia experiment (Fig. 2D), how can the authors be certain the defect is in cytokinesis, as opposed to karyokinesis or due to endoreduplication?
5. For contrast with invasive areas of tumor growth, it would be helpful to provide CAIX staining of the main (non-invasive) tumor as a negative control, for example in the section shown in figure 3B.

6. In Figure 4G, what accounts for the difference between control (almost 25% with bone metastasis) and normoxia cells (0% bone metastasis)?
7. For CHO cell transfectants (Fig. S5), the EdU staining is perhaps not surprising. It is not so informative to see only the Y5R-EGFP transfectants. First, do cells transfected with other receptors (Y1, Y2 etc) not proliferate at all? Second, it would be helpful to show the GFP channel alongside the RFP channel used to document EdU positivity.
8. RhoA inhibitor only partly rescues the % of multinucleated cells in the presence of Y5R (fig. 6D). Yet the authors make the strong statement that “cytokinesis defects triggered by Y5R overexpression are the result of RhoA activation” (p. 10). This statement should perhaps be qualified given the incomplete rescue.
9. In Figure 7C, the meaning of “a trend toward significant difference” is not clear. Either the difference meets the pre-established threshold for significance, or it does not (in this case it does not).
10. In Fig. 7E, what accounts for the absence of any osseous metastases in the control groups (compare to figure 1)?
11. Figure 8Ai demonstrates that tumor cells at the border of necrotic areas had enlarged nuclei, which is very interesting. However there are also possible nonspecific effects due to tissue degeneration and cellular swelling that can be associated with necrosis. Can the authors please assess total cell size and not just nuclear size under these conditions?

Reviewer #3 (Remarks to the Author):

The manuscript entitled ‘Hypoxia-activated neuropeptide Y/Y5 receptor/RhoA pathway triggers chromosomal instability and bone metastasis in Ewing sarcoma.’ by Lu et al. investigates the role of hypoxia in Ewing sarcoma (ES) bone metastasis with both mouse models and in vitro system. They showed that hypoxia induced by femoral artery ligation promoted ES bone metastasis with the increase of hypertrophic cells having enlarged nuclei and cells of aneuploidy. In vitro, hypoxia also increased the fraction of cells with enlarged nuclei. Cells derived from polyploid ES cells pretreated with hypoxia displayed higher metastatic capability than those from cells under normoxia condition. Then Lu, et al. proposed a model that hypoxia-induced Y5R activation contributes to the aneuploid/chromosomal instability through RhoA over-activation.

Generally, this manuscript proposed an interesting model and connected hypoxia, chromosome instability and neuropeptide signaling. However, there are some major issues that should be resolved before further consideration of publication.

1. The author used the term of ‘chromosomal instability’ in the title and introduction, but in the main text, the major readout of hypoxia-induced sub-cellular abnormality is the enlarged nuclei and FISH. FISH is a direct readout of aneuploidy and enlarged nuclei is a phenomenon associated with aneuploidy. Aneuploidy is a snapshot of certain chromosomal status but chromosomal instability is a turn to describe the dynamics of chromosome status. Although high chromosomal instability can lead to increased levels of aneuploidy occurrence, it is not necessarily required. To demonstrate the chromosomal instability is altered upon hypoxia, chromosomal missegregation during mitosis (lagging

chromosomes, chromosomal bridges, etc) should be scored.

2. While clarifying whether and how Y5R mediates signaling through hypoxia to chromosomal instability, the authors showed that the Y5R antagonist partially suppressed the hypoxia-induced nuclei enlargement (Fig 6G). Given that the mechanistic relationship between the aneuploidy and enlarged nuclei is still unclear, more direct assay illustrating the aneuploidy (like FISH used somewhere else in the manuscript) should be used here.

3. To demonstrate that Y5R promotes bone metastasis by sensing hypoxia condition, the author showed that KO Y5R specifically affected the bone metastasis in the FAL group but not the control group (Fig 7E). However, the metastasis number in Control -Dox here was almost zero and leaves no dynamic range for analysis. Metastasis number here in Control -Dox (~0) does not match the result in Fig 1D. Please clarify.

4. Nuclear area was extensively been used through the manuscript, please clarify how many nuclei from how many tumors from how many mice were used for each violin plots in supplementary data to show the in between tumor- and between mice variations. Sometimes, pooling all the data together might not be the best way for statistical analysis. Also hypoxia are not expected to be equally distributed within the tumor, does the pattern of tumor cells with enlarged nuclei correlate with that of hypoxia levels.

Jun Li

We thank the reviewers for the thorough evaluation of our manuscript and constructive comments. Per the reviewers' suggestions, we have included additional analyses and data to the revised version of this manuscript. Please see each point below (in grey) with our responses. Major changes to the manuscript are highlighted in yellow within the document. We believe that the inclusion of additional data and clarifications suggested by the review panel significantly improved our manuscript, and we thank the reviewers for the constructive comments.

Reviewer #1 (Remarks to the Author):

The manuscript from Lu et al. describes a study in which the role of NPY receptors in hypoxia-mediated metastasis in Ewing's Sarcoma (ES) is investigated. The authors show data that link hypoxia to aneuploidy and CIN, which is mechanistically linked to NPY receptor activity in in vitro studies, before confirming a role for NPY receptors in metastasis in their in vivo model of ES. This study builds on previous data showing hypoxia-driven genomic instability, including in particular a seminal study from the Bristow research group (Nature Genetics 2019) in which this is described in multiple tumour types.

As the authors point out, the effects of hypoxia in ES both in vitro and in vivo are relatively well described, although specific links between hypoxia and CIN in this cancer are not yet characterised. The highly novel aspect of this study is the link between hypoxia, NPY signalling, aneuploidy and metastasis. Overall, this is an interesting study, much of which has been carried out to a very high standard. However, there are some gaps in the evidence that in my view need to be addressed in order for the proposed mechanism to be entirely convincing.

A very general point is that how hypoxia in vitro was elicited in vitro is inadequately described, with the description of a 'chamber' at 0.1% O₂ (in materials and methods) an insufficient level of detail to understand whether this system was artefact-free. Studies of the effects of hypoxia on any aspect of cell phenotype where oxidative stress could play a role require particularly close attention paid to the system in use. As the equipment isn't detailed, we cannot assess whether 0.1% O₂ is an accurate, and more importantly a maintained level of oxygen tension over the timecourse of the experiment (72h of 0.1% O₂ as described for Figure 2D-F). Given that cells in hypoxia use oxygen, the oxygen tension would need to be actively monitored and 'topped-up' to avoid anoxia.

Response: In our study, hypoxia was created in an O₂ Control InVitro Cabinet (COY Lab Products, Grass Lake, MI) equipped with an oxygen sensor regulating a flow of 95% N₂ and 5% CO₂ gas mixture to maintain the desired O₂ levels throughout the duration of the experiment. A detailed description of the hypoxia chamber used has been added to the method section (page 14, lines 387-390).

Over 72h of severe hypoxia, significant amounts of other nutrients are used from the medium – most importantly perhaps the glucose, which could well be exhausted within 48h or less. This is known to increase CIN, and therefore measurements of glucose (at least) in the medium at the end of the experiment would be important to demonstrate that hypoxia alone resulted in the phenotypes observed – given that the in vivo model was also hyponutrientic (i.e. ischemic).

Response: Per reviewer's recommendation, we performed glucose measurements in the conditioned media upon 72h culture in normoxia or hypoxia. In both conditions, glucose remained readily available in the cell culture media throughout the duration of the experiment (approx. 3 mg/ml). There was also no significant difference in glucose levels in media from normoxic and hypoxic cultures. This data has been included in the Results section (page 6, lines 122-124), Methods section (page17-18, lines 497-513) and Figure S3c.

Figure 1G – an average number of around 0.3-0.4 bone metastases/animal is shown, while in Figure 1H around 30% of animals were suggested to have bone metastases. This suggests that around 6 animals had a single detectable metastasis, given an n=19. It isn't clear from the materials and methods how thoroughly the bones were checked for metastases, and the results could therefore show a higher rate of metastatic growth within the bone, rather than the metastatic spread induced by aneuploidy. This is an important point as the results may therefore reflect cell proliferation in the metastases rather than the process of metastasis.

Response: We thank the reviewer for noting the lack of clarity in our description of this analysis. Bone metastasis was assessed in all mice from each experimental group according to the same standardized protocol involving a combination of imaging and histopathological analysis:

- 1) Whole body MRI scans were analyzed for the presence of bone abnormalities indicative of potential metastatic spread. Any affected bones were collected during necropsy, sectioned and stained with H&E to confirm the presence of the tumor and therefore validate the MRI findings. In case of large lesions, longitudinal MRI allowed for distinguishing true osseous metastases initiated in the bone from those originating in soft tissues that subsequently invaded the neighboring bones.
- 2) Independently of the MRI analysis, all long bones and vertebral columns were collected as the most commonly affected bones in our model. These bones were subsequently cut longitudinally to expose the bone marrow cavity for the entire length of the bone, then sectioned and stained with H&E. These sections were analyzed by an experienced pathologist for the presence of bone metastases. This approach allowed us to detect small bone lesions at the initial stages of development, which were below the MRI detection level. Please see examples in the MS Review Figure 1.

The detailed description of the above procedure was added to the Methods section (page 16, lines 441-447).

MS Review Figure 1: Examples of osseous micrometastases detected by histopathology in mice bearing ES xenografts

To further exclude the possibility that the observed differences in the number of metastases were due to the differences in proliferation levels, we performed Ki67 staining in tissues from bone and extraosseous metastases developing in control and FAL-treated mice. As shown in MS Review Figure 2, the proliferation rates were comparable in control and hypoxic groups from both SK-ES-1 and TC71 xenografts.

MS Review Figure 2: Proliferation rates are comparable in metastases from control and FAL-treated mice

Representative images of extraosseous and bone metastatic tissues derived from control and FAL-treated mice and immunostained for a proliferation marker, Ki67. Scale bar: 50µm.

In addition, Ki67 immunostaining revealed the presence of enlarged, proliferative cells in hypoxic areas of primary tumors and bone metastases developing upon FAL. This new data has been included in the Results section (page 9-10, line 248-251) and Figure S13.

In Supplementary Figure 4, the authors show changes in cell viability in normoxia and hypoxia after doxorubicin treatment. Firstly, cross-comparison of normoxia and hypoxia is required in order to appreciate how hypoxia alters therapy response, and how the ploidy affected this. Secondly, as MTS

is based on cell redox, the signal for the same cell in hypoxia will be altered, so is not a suitable assay to use for hypoxia studies.

Response: We thank the reviewer for pointing out this inconsistency in our data. To address the above concerns, we assessed the survival of ES cells upon doxorubicin treatment by counting viable cells (trypan blue exclusion). While this experiment confirmed that HYP-4n cells are more resistant to chemotherapy, the response was comparable in both normoxia and hypoxia (MS Review Figure 3). As suggested by the reviewer, these data indicate that differences we observed between normoxia and hypoxia using an MTS assay reflect alterations in metabolism of HYP-4n cells rather than their increased chemoresistance under hypoxic conditions. Since the detailed characteristics of the HYP-4n cell metabolism is beyond the scope of the current manuscript, we removed the data on the chemoresistance in hypoxia.

The CHO model used to isolate the effect of Y5R expression on rate of aneuploidy also requires some clarification. It isn't clear that CHO cells expression NPY for autocrine/paracrine activation of the exogenously-expressed receptor. If not, how are the authors proposing that the receptor is activated in this system.

Response: The lack of the endogenous NPY receptor expression is a unique feature of CHO-K1 cells, making them an ideal model to test Y5R functions without interactions with other types of NPY receptors (Abuhsaud et al., 2020; Czarnecka et al., 2019). In addition, CHO-K1 cells do not express endogenous NPY, as we have confirmed by RT-PCR and ELISA (Czarnecka et al., 2019). However, as a peptide circulating in the blood, NPY is always present in cell culture media supplemented with FBS. This amount (typically ranging from $1 \times 10^{-11}\text{M}$ – $1 \times 10^{-10}\text{M}$) is in many of our systems sufficient to activate its receptors. To further prove that this effect is NPY-mediated, we performed a parallel experiment in 0.1% FBS culture medium, which has reduced levels of serum NPY. Under these conditions, the number of multinucleated CHO-K1/Y5R-EGFP cells was comparable to the CHO-K1/EGFP control, and addition of exogenous NPY significantly increased their frequency, mimicking the effect of 10% FBS. This data was described in the Results section (page 8, lines 203-210) and added as Figure 6d.

The microscopy images shown are not of sufficient resolution to confirm cell surface expression of Y5R – it is clearly key that the receptor is appropriately sited in this system. Indeed, it appears that expression is often localised to other intracellular structures (Figure 5A), and perhaps even the mitotic machinery.

Response: To confirm cell surface expression of Y5R, we used confocal microscopy to demonstrate co-localization of the Y5R-EGFP fusion protein and wheat germ agglutinin (WGA), which serves as a plasma membrane marker. Representative images have been added to the revised manuscript (Results section, page 7, lines 171-172 and Figure S7b). A detailed characterization of the subcellular localization of Y5R in the CHO-K1/Y5R-EGFP cells can be found in our previously published papers (Abualsaud et al., 2020; Czarnecka et al., 2019). We have used both confocal microscopy and treatment with membrane impermeable crosslinkers to demonstrate the membrane localization of the peptide and have shown a dynamic nature of Y5R-EGFP fusion protein localization. While a fraction of Y5Rs is localized to specific areas of the cell membrane, there is also a large population of the receptor present in intracellular vesicles. This pattern changes, depending on the presence of the ligand, as the receptor is rapidly internalized after its stimulation. Moreover, the membrane localization of the Y5R is observed in the leading edges of migrating cells and on the outer edges of cell colonies.

In Supplementary Figure 6, where the effect of the Rho inhibitor on cellular DNA content was observed, the traces must be overlaid so that a comparison can be made by the reader. As it stands, the non-quantified axis does not allow the comparison. As another comment, is peak shift the most appropriate measure to use? As it stands, the conclusion might be that the Rho inhibitor reduces the 4c peak to more like a 3.8 peak (although it isn't possible to tell what the result of the Rho inhibitor is with the axes presented) – this isn't necessarily the outcome required, given that 3.8 is still substantially more than 2c.

Response: The shift in 4c peak position that was observed in CHO-K1/Y5R-EGFP cells is consistent with the beginning of the next cell cycle (i.e. DNA synthesis) of tetraploid cells arising due to the Y5R-induced cytokinesis failure. No such shift in the position of 4c peak in Y5R-transfected vs non-transfected cells was observed in the presence of Rho inhibitor. However, we agree with the reviewer that flow cytometry is not the best way of measuring such changes, as differences in levels of cell proliferation and death between experimental groups interfere with data analysis. Hence, we removed the flow cytometry data previously shown in Figure S6. Instead, we focused on using microscopic images to analyze the percentage of multinucleated cells as a more direct measure of the frequency of cytokinesis defects and updated our analysis in Figure 6d.

In Figure 7, an elegant system in which Dox-inducible CRISPR-mediated Y5R mutation is utilised in vivo to confirm the mechanism by which hypoxia leads to aneuploidy and metastasis. Figure 7F requires a statistical comparison between Dox- and Dox+ in order to confirm that loss of Y5R alters rate of bone metastasis.

Response: The difference between -Dox and +Dox FAL groups shown in Figure 7f does not reach statistical significance ($p=0.187$). Nevertheless, unlike in -Dox group, the difference between the frequency of bone metastases between control and FAL-treated mice in +Dox group was also not statistically significant, indicating that Y5R gene editing averted the increase in bone metastasis induced by tumor hypoxia. Importantly, as mentioned in the Results section (page 9, lines 235-239), bone metastases were observed in only one mouse in +Dox group. Out of two bone metastases in total detected in this animal, one developed from a clone with an intact Y5R sequence. It is plausible that the same escape mechanism drove the development of the second metastasis in this mouse. However,

we did not have sufficient material to perform sequencing and prove this directly. Therefore, this metastasis was still included in the analysis.

Reviewer #2 (Remarks to the Author):

Review of Lu et al, "Hypoxia-activated neuropeptide Y/Y5 receptor/RhoA pathway triggers chromosomal instability and bone metastasis in Ewing sarcoma". In this manuscript, the investigators examine the links between tumor hypoxia, chromosomal instability and osseous metastasis in Ewing sarcoma, building on the group's previous work examining the role of NPY signaling in the disease. They use a variety of in vitro and in vivo approaches to dissect the contribution of hypoxia and to establish a role for the Y5 neuropeptide receptor and RhoA signaling in altering cell division properties, leading to cells with >4c DNA content and a propensity for bony metastasis. A particularly strong piece of data is the ability of in vivo CRISPR editing of Y5R to prevent osseous metastasis in xenograft models. Overall the study is very well-done and represents an important advance for the field. There are some areas of ambiguity that should be addressed to increase the impact of the manuscript.

Specific comments:

1. How was 3 days of time between FAL and leg amputation arrived at as an experimental timepoint?

Response: The FAL method has been described in detail in our previous paper (Tilan et al., 2013). We have shown that upon initial ischemia following FAL, blood flow in the affected leg is restored within three days due to collateral vessel remodeling and angiogenesis. These findings were used to determine the three-day experimental timepoint for subsequent leg amputation. In our ES hypoxia experiments, this time period allowed for tumor revascularization, and thereby facilitated an escape of the metastatic tumor cells from the tumor exposed to hypoxia. This approach recapitulates transient hypoxia and reoxygenation events known to occur in solid tumors. The details of the ES hypoxia model *in vivo* can be found in our methods paper (Hong et al., 2016).

2. If large tumors exhibit high endogenous hypoxia, do they give rise to more osseous metastasis?

Response: Indeed, the frequency of metastases in mice bearing larger ES tumors (1cm³) is higher than in those with smaller tumors (0.25cm³). This includes both bone and extraosseous metastases. However, the direct comparison between these tumors cannot be made, as the time required to reach this large tumor volume is also significantly longer.

3. For the analyses in Figure 2, please provide more detail on the experimental and statistical design. How many nuclei were measured for the analyses in Figures 2A and 2D? (Consider providing the data points as part of the violin plots). Please provide n of the number of cells scored for bar charts in Figures 2B, 2C, 2E (methods section for FISH mentions at least 100 nuclei scored, was this also the case for scoring >4c DNA content, or was that based on integration of FACS plot?)

Response: The n pertaining to numbers of scored nuclei has been added to all figure legends describing nuclear size analysis or FISH data. In addition, to provide a better illustration of the variability between tumors, we show the nuclear sizes separately in each analyzed tumor, as newly included

graphs (Figures S2a-b, S4b-c, S10c-d). The graphs pertaining to the >4c DNA content were based on the FACS analysis.

4. In the in vitro hypoxia experiment (Fig. 2D), how can the authors be certain the defect is in cytokinesis, as opposed to karyokinesis or due to endoreduplication?

Response: The immediate consequence of cytokinesis failure is the formation of cells with two nuclei that can subsequently fuse and form one large tetraploid nucleus. This phenomenon was directly observed in CHO-K1/Y5R-EGFP cells by time-lapse microscopy and in fixed cells 24h post-transfection (Figures 5d-e, 6d). Similarly, we have observed a significant increase in the frequency of multinucleated ES cells exposed to hypoxia for 24h, which was followed by the formation of cells with hypertrophic nuclei. As cell fusion can be another mechanism leading to the formation of multinucleated cells, we performed a parallel experiment using co-culture of ES cells transfected with GFP or mCherry fluorescent proteins. Upon 24h of hypoxia exposure, the resulting multinucleated cells were stained exclusively with one of the above fluorescent proteins and no GFP and mCherry co-staining was observed in these cells, thereby excluding the contribution of cell fusions to their formation. This data confirmed that exposure to hypoxia leads to cytokinesis defects in ES cells and it has been included in the revised version of the manuscript (Results section, page 5, lines 111-117 and Figures 2f, S3a):

5. For contrast with invasive areas of tumor growth, it would be helpful to provide CAIX staining of the main (non-invasive) tumor as a negative control, for example in the section shown in figure 3B.

Response: CAIX is a hypoxia-inducible protein responsible for maintaining cytosolic pH under low oxygen conditions and is used as a marker of a long-term adaptation to hypoxia. Consequently, its immunostaining pattern in large solid tumors is similar to that of hypoxypromote-1 (HP-1), pimonidazole. Low-magnification images demonstrating similarities in the immunostaining pattern for CAIX and HP-1 were added to the revised manuscript as Figure S4a. The viable tumor areas, such as that shown in Figure 3b, are negative for both markers, while tissue surrounding necrotic areas, where hypoxic, yet still-alive cells reside, is highly positive. The greater extent of the CAIX staining is associated with the fact that HP-1 requires lower oxygen tension to bind to the cancer cells than this needed to induce CAIX expression.

6. In Figure 4G, what accounts for the difference between control (almost 25% with bone metastasis) and normoxia cells (0% bone metastasis)?

Response: In Figure 4G, the differences between control cells and normoxic cell fractions are not statistically significant ($p = 0.4965$ and $p = 0.5091$ by Fisher's exact test for 2n and 4n cells, respectively) and can reflect test variability, which is typically high for metastatic models. Moreover, the NOR-2n cell fraction is devoid of polyploid cells, while NOR-4n tumors did not metastasize at all. Nevertheless, the robust increase in the frequency of bone metastases in mice bearing HYP~4n cells clearly identifies this fraction as the one responsible for osseous dissemination, which was the main outcome of this experiment.

7. For CHO cell transfectants (Fig. S5), the EdU staining is perhaps not surprising. It is not so informative to see only the Y5R-EGFP transfectants. First, do cells transfected with other receptors

(Y1, Y2 etc) not proliferate at all? Second, it would be helpful to show the GFP channel alongside the RFP channel used to document EdU positivity.

Response: Aside from polyploidy, the increase in cellular and nuclear size can be caused by cell death-related swelling or senescence. Thus, the purpose of showing a positive EdU staining in the enlarged CHO-K1/Y5R-EGFP transfectants was to provide evidence for their viability and ability to proliferate. As we have previously described, both Y1R and Y2R transfectants proliferate and their rate of proliferation increases upon NPY stimulation (Czarnecka et al., 2019). Thus, while the mitogenic capabilities of Y5R transfectants are not surprising, in our opinion the EdU staining is an important indication of their viability. To better illustrate this point, we included updated images as Figure S7c, which show GFP and RFP channels in CHO-K1 cells transiently transfected with Y5R-EGFP.

8. RhoA inhibitor only partly rescues the % of multinucleated cells in the presence of Y5R (fig. 6D). Yet the authors make the strong statement that “cytokinesis defects triggered by Y5R overexpression are the result of RhoA activation” (p. 10). This statement should perhaps be qualified given the incomplete rescue.

Response: We thank the reviewer for this important comment that revealed a potential pitfall in our previous research design. RhoA inhibition is technically challenging. As a basic cytoskeleton regulator, RhoA is involved in many cellular functions. In early stages of cytokinesis, RhoA activity is essential for the cleavage furrow ingression, while in the late stages it has to be inactivated for daughter cell detachment (Figure 6a). Our time lapse microscopy clearly demonstrated that Y5R overexpression causes defects in the abscission phase of cytokinesis (Figure 5d-e, Supplementary Movies). Nevertheless, both too-low and too-high RhoA activity results in cytokinesis failure by inhibiting its early or late stages, respectively, and leads to the same phenotype – multinucleated cells. We have observed that the Rho inhibitor at a previously used concentration (0.1µg/ml) blocks cytokinesis by itself, as evidenced by an increase in G2/M peak (Figure S8a). This phenomenon interfered with a proper analysis of our previous data - in addition to inhibiting Y5R-induced formation of polyploid cells, the inhibitor caused the same phenotype by interfering the cleavage furrow formation. To overcome this challenge, we decreased the concentration of the Rho inhibitor to 0.01µg/ml, which had no effect on basal proliferation, yet inhibited Y5R-mediated RhoA activation and fully prevented the formation of multinucleated cells. The data in Figure 6d has been updated accordingly, and the additional information on the effect of Rho inhibitor has been included in Figure S8.

9. In Figure 7C, the meaning of “a trend toward significant difference” is not clear. Either the difference meets the pre-established threshold for significance, or it does not (in this case it does not).

Response: The description of the data presented in Figure 7c was revised as follows (page 9, lines 225):

“While this treatment did not affect primary tumor growth (Figure S9e) or extra-osseous metastases, it decreased the percentage of mice with secondary bone lesions and the number of osseous metastases, although the latter difference did not reach statistical significance (Figure 7b-d).”

10. In Fig. 7E, what accounts for the absence of any osseous metastases in the control groups (compare to figure 1)?

Response: Metastasis is a complex process and can be affected by a variety of factors. Hence, we often observe differences in baseline metastasis levels between experiments. Moreover, in our experience, the expression of any viral vectors in ES cells changes their metastatic pattern, typically inhibiting cancer cell spread to some degree. We have observed this phenomenon upon transduction with a variety of vectors, including lentiviruses expressing Cas9, sgRNAs, shRNAs, cDNAs and their corresponding control vectors. To account for these effects, we used a Dox-inducible CRISPR/Cas9 system. This approach allowed us to inject all animals with the same cells and ensure that the observed differences are due to the Dox-induced gene editing rather than cell line-to-cell line variability. Consequently, even though the baseline is different, FAL in the control group exhibits the same effects as in the wild type cells.

11. Figure 8Ai demonstrates that tumor cells at the border of necrotic areas had enlarged nuclei, which is very interesting. However there are also possible nonspecific effects due to tissue degeneration and cellular swelling that can be associated with necrosis. Can the authors please assess total cell size and not just nuclear size under these conditions?

Response: We agree with the reviewer that the increase in nuclear size can also be associated with cellular swelling, a sign of cell death. Both polyploidy and cellular swelling lead to the increase in nuclear volume and overall cell size. However, cellular swelling is associated with characteristic changes within the cytoplasm and in the nucleus indicative of cell death, i.e. nuclear edema and/or pyknosis. In our analysis, we focused on enlarged cells without the above-mentioned degenerative changes (MS Review Figure 4). This information has been added to the Methods section (page 16, lines 453-460).

MS Review Figure 4: Examples of viable hypertrophic ES cells

ES xenograft tissue stained with H&E. The viable cells that are located at the edge of the necrotic area that were used for analyses are labeled with red asterisks. N - necrosis. Scale bar: 25 μ m.

Moreover, many enlarged cells identified by us at the edge of necrosis were positive for Ki67, a marker of active proliferation (Figure S13a). Numerous enlarged cells with strong Ki67 staining were also observed in FAL-exposed hypoxic primary tumors and their corresponding bone metastases (Figure S13b-d). In line with these findings, our in vitro experiments revealed that upon 72h exposure to severe hypoxia, the population of proliferating cells is limited to those with signs of hypertrophy (Figure S3b). These data have been added to the manuscript (Results section, page 5-6, lines 119-122 and Figure S3b; Results section, page 9-10, lines 248-251 and Figure S13).

Reviewer #3 (Remarks to the Author):

The manuscript entitled 'Hypoxia-activated neuropeptide Y/Y5 receptor/RhoA pathway triggers chromosomal instability and bone metastasis in Ewing sarcoma.' by Lu et al. investigates the role of hypoxia in Ewing sarcoma (ES) bone metastasis with both mouse models and in vitro system. They showed that hypoxia induced by femoral artery ligation promoted ES bone metastasis with the increase of hypertrophic cells having enlarged nuclei and cells of aneuploidy. In vitro, hypoxia also increased the fraction of cells with enlarged nuclei. Cells derived from polyploid ES cells pretreated with hypoxia displayed higher metastatic capability than those from cells under normoxia condition. Then Lu, et al. proposed a model that hypoxia-induced Y5R activation contributes to the aneuploid/chromosomal instability through RhoA over-activation.

Generally, this manuscript proposed an interesting model and connected hypoxia, chromosome instability and neuropeptide signaling. However, there are some major issues that should be resolved before further consideration of publication.

1. The author used the term of 'chromosomal instability' in the title and introduction, but in the main text, the major readout of hypoxia-induced sub-cellular abnormality is the enlarged nuclei and FISH. FISH is a direct readout of aneuploidy and enlarged nuclei is a phenomenon associated with aneuploidy. Aneuploidy is a snapshot of certain chromosomal status but chromosomal instability is a turn to describe the dynamics of chromosome status. Although high chromosomal instability can lead to increased levels of aneuploidy occurrence, it is not necessarily required. To demonstrate the chromosomal instability is altered upon hypoxia, chromosomal missegregation during mitosis (lagging chromosomes, chromosomal bridges, etc.) should be scored.

Response: We thank the reviewer for this important clarification. As suggested, we provided direct evidence for the hypoxia-induced increase in chromosomal instability by quantifying the frequency of mitotic segregation errors (misaligned and lagging chromosomes, anaphase/telophase bridges) and interphase abnormalities that are also considered as markers of chromosomal instability (micronuclei and nuclear blebs) in the following samples:

- 1) Primary cell cultures isolated from control and hypoxic tumors (Figures 2d, S2e-f)
- 2) ES cells exposed to hypoxia for 72h followed by a 24h incubation in normoxia to stimulate subsequent cell divisions (Figure 2i)
- 3) HYP~4n cells (Figure 4c)

2. While clarifying whether and how Y5R mediates signaling through hypoxia to chromosomal instability, the authors showed that the Y5R antagonist partially suppressed the hypoxia-induced nuclei enlargement (Fig 6G). Given that the mechanistic relationship between the aneuploidy and enlarged nuclei is still unclear, more direct assay illustrating the aneuploidy (like FISH used somewhere else in the manuscript) should be used here.

Response: Per the reviewer's request, we performed FISH analysis to supplement the data shown in Figure 6g and provide further evidence supporting the role of Y5R in hypoxia-induced aneuploidy (Figure 6h, Figure S9).

3. *To demonstrate that Y5R promotes bone metastasis by sensing hypoxia condition, the author showed that KO Y5R specifically affected the bone metastasis in the FAL group but not the control group (Fig 7E). However, the metastasis number in Control -Dox here was almost zero and leaves no dynamic range for analysis. Metastasis number here in Control -Dox (~0) does not match the result in Fig 1D. Please clarify.*

Response: This point has been addressed in the response to comments from reviewer #2. Please see also below:

Metastasis is a complex process and can be affected by a variety of factors. Hence, we often observe differences in baseline metastasis levels between experiments. Moreover, in our experience, the expression of any viral vectors in ES cells changes their metastatic pattern, typically inhibiting cancer cell spread to some degree. We have observed this phenomenon upon transduction with a variety of vectors, including lentiviruses expressing Cas9, sgRNAs, shRNAs, cDNAs and their corresponding control vectors. To account for these effects, we used a Dox-inducible CRISPR/Cas9 system. This approach allowed us to inject all animals with the same cells and ensure that the observed differences are due to the Dox-induced gene editing rather than cell line-to-cell line variability. Consequently, even though the baseline is different, FAL in the control group exhibits the same effects as in the wild type cells.

4. *Nuclear area was extensively been used through the manuscript, please clarify how many nuclei from how many tumors from how many mice were used for each violin plots in supplementary data to show the in between tumor- and between mice variations. Sometimes, pooling all the data together might not be the best way for statistical analysis.*

Response: The n pertaining to numbers of scored nuclei were added to all figure legends describing nuclear size analysis. To provide a better illustration of the variability between tumors, we show the nuclear sizes separately in each analyzed tumor as updated graphs (Figure S2a-b, S4b-c, S10c-d).

Also hypoxia are not expected to be equally distributed within the tumor, does the pattern of tumor cells with enlarged nuclei correlate with that of hypoxia levels.

Response: In rapidly growing solid tumors, such as ES, hypoxic, but still viable cells are typically present on the edges of necrosis (Figures 8a, S4a). We have also identified hypoxic areas in the bone invasion niches (Figure 3a). As shown in Figures 3b-c, 8, 9 and S13a, these are the areas where cells with enlarged nuclei and chromosome gains accumulate.

References:

- Abualsaud, N., Caprio, L., Galli, S., Krawczyk, E., Alamri, L., Zhu, S., Gallicano, G. I., & Kitlinska, J. (2020). Neuropeptide Y/Y5 Receptor Pathway Stimulates Neuroblastoma Cell Motility Through RhoA Activation. *Front Cell Dev Biol*, 8, 627090. <https://doi.org/10.3389/fcell.2020.627090>
- Czarnecka, M., Lu, C., Pons, J., Maheswaran, I., Ciborowski, P., Zhang, L., Cheema, A., & Kitlinska, J. (2019). Neuropeptide Y receptor interactions regulate its mitogenic activity. *Neuropeptides*, 73, 11-24. [https://doi.org/S0143-4179\(18\)30096-9](https://doi.org/S0143-4179(18)30096-9) [pii] 10.1016/j.npep.2018.11.008
- Hong, S. H., Tilan, J., Galli, S., Acree, R., Connors, K., Mahajan, A., Wietlisbach, L., Polk, T., Izycka-Swieszewska, E., Lee, Y. C., Cavalli, L. R., Rodriguez, O., Albanese, C., & Kitlinska, J. (2016). In vivo model for testing effect of hypoxia on tumor metastasis. *J Vis Exp*.(118).
- Tilan, J. U., Everhart, L. M., Abe, K., Kuo-Bonde, L., Chalothorn, D., Kitlinska, J., Burnett, M. S., Epstein, S. E., Faber, J. E., & Zukowska, Z. (2013). Platelet neuropeptide Y is critical for ischemic revascularization in mice. *Faseb J*. <https://doi.org/fj.12-213546> [pii] 10.1096/fj.12-213546

REVIEWERS' COMMENTS

Reviewer #1 (Remarks to the Author):

My original appraisal of the importance of the manuscript remains the same - the standard of work is overall very high and the studies performed set out a very interesting novel story regarding the mechanism by which hypoxia leads to chromosomal instability in ES.

I'd like to thank the authors for addressing all of my concerns, and feel that the manuscript is now easier to follow, and importantly easier for other researchers to reproduce and follow-up in the future.

Reviewer #2 (Remarks to the Author):

The revised manuscript by Lu, Kitlinska and co-workers includes substantial new data in response to the points raised in review. The new data in Figure 6 and supplementary figures (esp S4 and S13) provide important clarification. Overall the authors have convincingly addressed the issues identified by the reviewers and the revised manuscript is suitable for publication.

Reviewer #3 (Remarks to the Author):

Thank you for great efforts to improve the manuscript. Additional data and/or explanation resolved most of my concerns except the following one which is critical to this manuscript.

One important novelty of this manuscript is that Y5R links hypoxia and chromosomal instability (CIN). The old version used 'enlarged nuclei' as the indicator of CIN throughout the paper.

I am glad that you were able to show the mitotic errors are higher in cells derived from hypoxia tumors than cells from control tumors. It might be unrealistic to score all the cells used in this manuscript. However, it would be necessary to show at least some cases that Y5R perturbation alters the CIN levels rather than the 'enlarged nuclei'.

For example, can you score the cells in Fig 6g?

SK-ES-1 cells in NOR or HYP for 72h, with or without Y5R antagonist.

This experiment would be a minimum effort but will be critical for the claim that "hypoxia-induced stimulation of the neuropeptide Y (NPY)/Y5 receptor (Y5R) pathway, which led to RhoA over-activation and cytokinesis failure." in the abstract.

We thank the reviewers once again for the thorough evaluation of our manuscript and positive comments. These constructive reviews were extremely helpful in identifying aspects of the initial version of our manuscript in need of improvement. Our revision of the paper based on these comments allowed us to significantly enhance its quality and increase its merit. In response to the request from Reviewer #3, we included additional data supporting the role of the Y5R pathway in hypoxia-induced chromosomal instability (please see below for details).

Reviewer #1 (Remarks to the Author):

My original appraisal of the importance of the manuscript remains the same - the standard of work is overall very high and the studies performed set out a very interesting novel story regarding the mechanism by which hypoxia leads to chromosomal instability in ES.

I'd like to thank the authors for addressing all of my concerns, and feel that the manuscript is now easier to follow, and importantly easier for other researchers to reproduce and follow-up in the future.

Reviewer #2 (Remarks to the Author):

The revised manuscript by Lu, Kitlinska and co-workers includes substantial new data in response to the points raised in review. The new data in Figure 6 and supplementary figures (esp S4 and S13) provide important clarification. Overall the authors have convincingly addressed the issues identified by the reviewers and the revised manuscript is suitable for publication.

Reviewer #3 (Remarks to the Author):

Thank you for great efforts to improve the manuscript. Additional data and/or explanation resolved most of my concerns except the following one which is critical to this manuscript.

One important novelty of this manuscript is that Y5R links hypoxia and chromosomal instability (CIN). The old version used 'enlarged nuclei' as the indicator of CIN throughout the paper. I am glad that you were able to show the mitotic errors are higher in cells derived from hypoxia tumors than cells from control tumors. It might be unrealistic to score all the cells used in this manuscript. However, it would be necessary to show at least some cases that Y5R perturbation alters the CIN levels rather than the 'enlarged nuclei'. For example, can you score the cells in Fig 6g?

SK-ES-1 cells in NOR or HYP for 72h, with or without Y5R antagonist.

This experiment would be a minimum effort but will be critical for the claim that "hypoxia-induced stimulation of the neuropeptide Y (NPY)/Y5 receptor (Y5R) pathway, which led to RhoA over-activation and cytokinesis failure." in the abstract.

We thank the reviewer for identifying this important gap in our data. As requested, we performed an additional quantification of mitotic segregation errors in the experimental set-up corresponding to that shown in Fig. 6g – SK-ES-1 cells subjected to hypoxia for 72h, with or without Y5R antagonist. The cells were subsequently subjected to media change and cultured in normoxia for 24h to stimulate proliferation. The results are presented in supplementary figure 9a.